# Non-invasive decision support for NSCLC treatment using PET/CT radiomics

Wei Mu [1,11], Lei Jiang[2,11], JianYuan Zhang [3,4], Yu Shi[1], Jhanelle E. Gray[5], Ilke Tunali[1], Chao Gao[6,7], Yingying Sun[6,7], Jie Tian [8,9], Xinming Zhao [3,12✉], Xilin Sun [6,7,12✉], Robert J. Gillies [1,12✉] & Matthew B. Schabath [5,10,12✉]

Two major treatment strategies employed in non-small cell lung cancer, NSCLC, are tyrosine kinase inhibitors, TKIs, and immune checkpoint inhibitors, ICIs. The choice of strategy is based on heterogeneous biomarkers that can dynamically change during therapy. Thus, there is a compelling need to identify comprehensive biomarkers that can be used longitudinally to help guide therapy choice. Herein, we report a [18]F-FDG-PET/CT-based deep learning model, which demonstrates high accuracy in *EGFR* mutation status prediction across patient cohorts from different institutions. A deep learning score (EGFR-DLS) was significantly and positively associated with longer progression free survival (PFS) in patients treated with EGFR-TKIs, while EGFR-DLS is significantly and negatively associated with higher durable clinical benefit, reduced hyperprogression, and longer PFS among patients treated with ICIs. Thus, the EGFR-DLS provides a non-invasive method for precise quantification of *EGFR* mutation status in NSCLC patients, which is promising to identify NSCLC patients sensitive to EGFR-TKI or ICI-treatments.

[1] Department of Cancer Physiology, H. Lee Moffitt Cancer Center and Research Institute, Tampa, FL, USA. [2] Department of Nuclear Medicine, Shanghai Pulmonary Hospital, Tongji University School of Medicine, Shanghai, China. [3] Department of Nuclear Medicine, the Fourth Hospital of Hebei Medical University, Hebei, China. [4] Department of Nuclear Medicine, Baoding No.1 Central Hospital, Baoding, Hebei, China. [5] Department of Thoracic Oncology, H. Lee Moffitt Cancer Center and Research Institute, Tampa, FL, USA. [6] NHC and CAMS Key Laboratory of Molecular Probe and Targeted Theranostics, Molecular Imaging Research Center (MIRC), Harbin Medical University, Harbin, Heilongjiang, China. [7] TOF—PET/CT/MR center, the Fourth Hospital of Harbin Medical University, Harbin Medical University, Harbin, Heilongjiang, China. [8] Beijing Advanced Innovation Center for Big Data-Based Precision Medicine, School of Medicine, Beihang University, Beijing, China. [9] CAS Key Laboratory of Molecular Imaging, Institute of Automation, Chinese Academy of Sciences, Beijing, China. [10] Department of Cancer Epidemiology, H. Lee Moffitt Cancer Center and Research Institute, Tampa, FL, USA. [11]These authors contributed equally: Wei Mu, Lei Jiang. [12]These authors jointly supervised this work: Xinming Zhao, Xilin Sun, Robert J. Gillies, Matthew B. Schabath. ✉email: xinm_zhao@163.com; sunxl@ems.hrbmu.edu.cn; Robert.Gillies@Moffitt.org; matthew.schabath@moffitt.org

Non-small cell lung cancer (NSCLC) is the most common histologic subtype of lung cancer and the leading cause of cancer-related death worldwide, with a dismal 5-year survival rate of 5% for the patients diagnosed with metastatic disease[1]. The emergence of two treatment paradigms has revolutionized cancer treatment and improved survival clinical among subsets of patients, with advanced NSCLC: targeted therapy represented by epidermal growth factor receptor (EGFR) tyrosine kinase inhibitors (TKIs)[2,3] and immune checkpoint inhibitors (ICIs) targeting the programmed death-1 (PD-1) receptor on T-cells, or the programmed death ligand-1 (PD-L1) expressed by tumor cells[4–8]. Patients are treated with TKIs when harboring an activating mutation of the EGFR, resulting in objective response rates (ORR) as high as 80% compared to an ORR of 10% in patients with wild-type EGFR[9]. Notably, patients with EGFR mutations have a low ORR to ICI treatments[10], and this is believed to be due to the lack of inflammatory microenvironment[11]. Therefore, an accurate estimation of EGFR mutation status could inform therapy choice between EGFR-TKI or ICI treatment, which would improve the patient outcome.

At present, EGFR mutation status[12] is determined by tissue-based assays, which have many limitations, including inter alia: sampling bias due to the heterogeneous nature of tumors, a requirement for invasive biopsies with associated morbidities, the assays are not rapid, can be expensive, and may fail to yield actionable results due to insufficient quantity or quality of the tissue[13]. Further, EGFR mutational status and immune landscape may change during the course of therapy and progression[14]. As such, high-throughput and, ideally, noninvasive longitudinal methods that can predict EGFR mutational status is a critical need. Recently, noninvasive molecular imaging with positron emission tomography (PET) with a radiotracer, [18]F-radiolabeled polyethylene glycol (PEG)–modified (PEGylated) anilinoquinazoline derivative, 2-(2-(2-(2-(4-(3-chloro-4-fluorophenylamino)-6-methoxyquinazolin-7-yl)oxy)ethoxy)ethoxy)ethoxy) ethyl 4-methylbenzenesulfonate ([18]F-MPG), has shown potential to detect mutated EGFR and identify patients who are likely to benefit from EGFR-TKI treatment[15]. However, this radiotracer is not routinely available.

In contrast to [18]F-MPG, PET/CT imaging of fluorodeoxyglucose, [18]F-FDG, is widely used for staging of patients with NSCLC. Further, uptake of this tracer is known to be affected by EGFR activation and inflammation[16]. Early studies have shown that radiomics features extracted from PET/CT images, and CT images can predict gene expression patterns and EGFR mutation status[17,18]. However, the hand-crafted radiomics features used in these prior studies were computed from a segmented tumor volume, which is dependent on precise tumor boundary delineation and does not consider information that may be present in the peritumoral microenvironment. Hence, advanced artificial intelligence models using deep learning approaches that do not require accurate segmentation have been investigated to achieve better performance in diagnosis, prediction, and prognosis[19,20].

In this study, we develop an [18]F-FDG PET/CT-based deep learning model, which could accurately classify EGFR mutation status, using two retrospective cohorts of patients accrued from two institutions: the Shanghai Pulmonary Hospital (SPH), Shanghai, China, and the Fourth Hospital of Hebei Medical University (HBMU), Hebei, China. To evaluate the performance of the EGFR prediction model, an external test cohort from the Fourth Hospital of Harbin Medical University (HMU), Harbin, China is used. Using the model generated deep learning score (EGFR-DLS), further evaluation of the potential value in guiding therapy choice is performed in the TKI-treated patients from HMU and ICI-treated patients from H. Lee Moffitt Cancer Center and Research Institute (HLM), Tampa, Florida, respectively (details shown in Fig. 1).

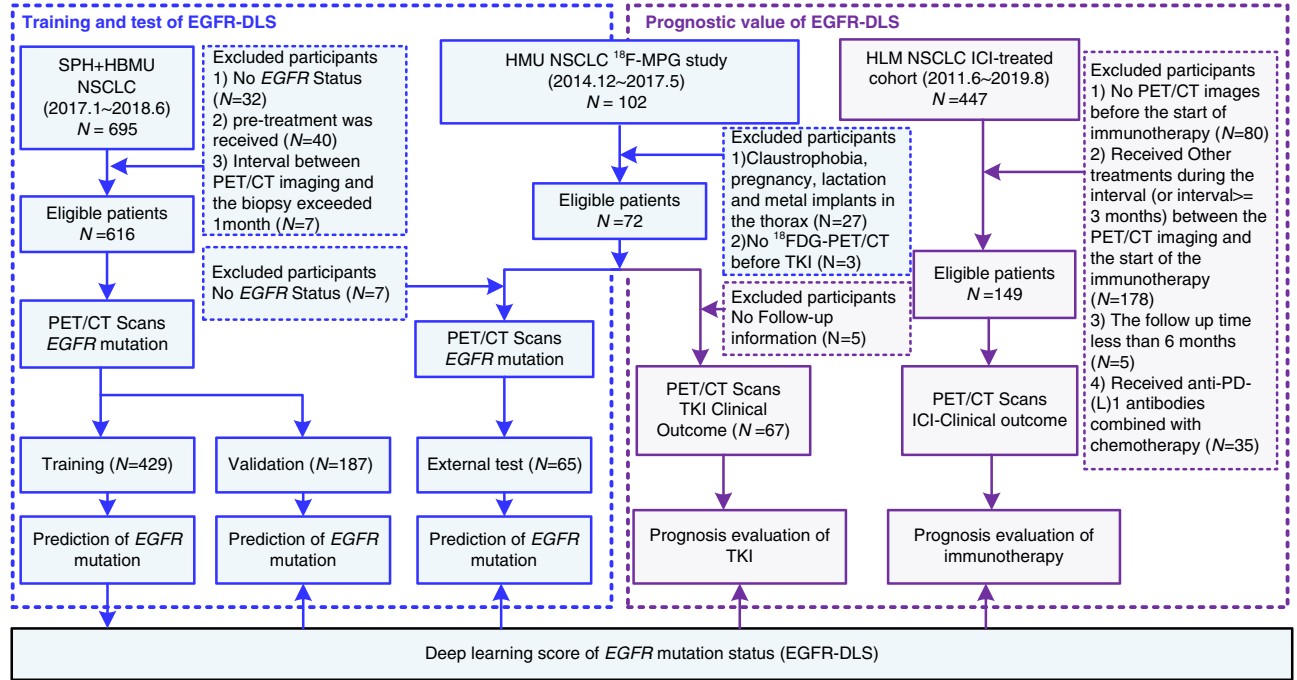

**Fig. 1 Study design and inclusion and exclusion diagram.** The SPH and HBMU data comprised EGFR mutation status and the corresponding imaging data, and was used to train and validate the deep learning score (EGFR-DLS) generated by the deep learning model. The HMU TKI-treated data comprised EGFR mutation status, and the corresponding imaging data was used for the external test of the EGFR-DLS and also used for the investigation of the prognostic value of the EGFR-DLS for TKI treatment. The HLM ICI-treated data comprised patients included in anti-PD-1 and anti-PD-L1 immunotherapy, and was used for the investigation of the prognostic value of the EGFR-DLS for immunotherapy.

**Table 1 Demographic and clinical characteristics of EGFR mutation related patients.**

| Characteristic | Training cohort (N = 429) | | | Validation cohort (N = 187) | | | HMU EGFR-test-cohort (N = 65) | | |
|---|---|---|---|---|---|---|---|---|---|
| | EGFR+ (N = 201) | EGFR− (N = 228) | P | EGFR+ (N = 75) | EGFR− (N = 112) | P | EGFR+ (N = 36) | EGFR− (N = 29) | P |
| Age(y) | | | 0.55 | | | 0.99 | | | 0.21 |
| Mean ± SD | 62.79 ± 8.65 | 63.26 ± 8.94 | | 62.6 ± 9.14 | 62.47 ± 9.32 | | 58.81 ± 10.4 | 61.07 ± 9.07 | |
| Sex, no. (%) | | | <.001* | | | <.001* | | | .047* |
| Female | 137 (68.16) | 57 (25) | | 50 (66.67) | 23 (20.54) | | 24 (66.67) | 12 (41.38) | |
| Male | 64 (31.84) | 171 (75) | | 25 (33.33) | 89 (79.46) | | 12 (33.33) | 17 (58.62) | |
| TNM stage | | | 0.23 | | | <.001* | | | 0.52 |
| I | 116 (57.71) | 111 (48.68) | | 47 (62.67) | 56 (50) | | 8 (22.22) | 7 (24.14) | |
| II | 30 (14.93) | 47 (20.61) | | 8 (10.67) | 24 (21.43) | | 2 (5.56) | 1 (3.45) | |
| III | 30 (14.93) | 42 (18.42) | | 8 (10.67) | 20 (17.86) | | 6 (16.67) | 9 (31.03) | |
| IV | 25 (12.44) | 28 (12.28) | | 12 (16) | 12 (10.71) | | 20 (55.56) | 12 (41.38) | |
| Histology (baseline), no. (%) | | | <.001* | | | <.001* | | | .002* |
| ADC | 197 (98.01) | 157 (68.86) | | 75 (100) | 80 (71.43) | | 36 (100) | 22 (75.86) | |
| SCC | 4 (1.99) | 71 (31.14) | | 0 (0) | 31 (27.68) | | 0 (0) | 7 (24.14) | |
| Smoking status, no. (%) | | | <.001* | | | 0.076 | | | .070 |
| Never | 156 (77.61) | 80 (35.09) | | 56 (74.67) | 36 (32.14) | | 25 (69.44) | 13 (44.83) | |
| Former | 45 (22.39) | 148 (64.91) | | 19 (25.33) | 76 (67.86) | | 11 (30.56) | 16 (55.17) | |
| SUV$_{max}$ | | | <.001* | | | <.001* | | | 0.99 |
| Mean ± SD | 7.92 ± 5.22 | 10.17 ± 5.7 | | 6.97 ± 4.82 | 10.32 ± 5.52 | | 9.47 ± 7.67 | 8.41 ± 6.06 | |
| PD-L1 status, no. (%) | | | .069 | | | 0.28 | | | NA |
| ≥50% | 10 (4.98) | 24 (10.53) | | 5 (6.67) | 12 (10.71) | | NA | | |
| 1–49% | 12 (6.97) | 30 (13.16) | | 3 (4.00) | 12 (10.71) | | | | |
| 0% | 89 (43.28) | 102 (44.74) | | 32 (42.67) | 48 (42.86) | | | | |
| Unknown | 90 (44.78) | 72 (31.58) | | 35 (46.67) | 40 (35.71) | | | | |
| EGFR-deep learning score (EGFR-DLS) | | | <.001* | | | <.001* | | | <.001* |
| Median | 0.65 | 0.34 | | 0.63 | 0.38 | | 0.55 | 0.26 | |
| (IQR) | (0.54, 0.73) | (0.15, 0.34) | | (0.55, 0.70) | (0.21, 0.49) | | (0.38, 0.66) | (0.08, 0.44) | |

The comparison of age, EGFR-DLS, and SUV$_{max}$ between two groups was performed with two-sided Wilcoxon sign-rank test, and the rest variables were compared with two-sided Fisher's test. IQR is short for interquartile range. ADC is short for adenocarcinoma and SCC is short for squamous cell carcinoma.
NA not available.
*p Value <0.05.

## Results

**Demographic and clinical characteristics.** Table 1 shows the demographic and clinical characteristics of the patients used to train and test EGFR-DLS, as a potential diagnostic biomarker for *EGFR* mutation status. The prevalence of mutant *EGFR* in the training, validation, and HMU *EGFR* test cohorts was 46.85%, 40.11%, and 55.38%, respectively. There was no significant difference for histology ($p = 0.26$) or smoking status ($p = 0.19$) among these three cohorts, but the prevalence of females was significantly higher ($p = 0.033$) in the HMU cohort. The demographic and clinical characteristics of the patients to test the utility of EGFR-DLS to predict response are presented in Table 2. For the EGFR-TKI-treated cohort ($n = 67$), the median progression-free survival (PFS) was 6.1 months, with 27 (40.30%), 9 (13.43%), and 31 (46.27%) patients responding with progressive disease (PD), stable disease (SD), and complete response (CR)/ partial response (PR), respectively. The ICI-treated cohort ($n = 149$) had a median PFS of 7.67 months, with 31 (20.81%) and 87 (58.39%) patients responding with hyperprogression (time-to-treatment failure (TTF) < 2 months) and durable clinical benefit (DCB, PFS ≥ 6 months).

**Performance of EGFR-DLS in predicting EGFR mutation status.** To discriminate *EGFR*-mutant type from wild type, the EGFR-DLS yielded area under the receiver operating characteristics curves (AUCs) of 0.86, 0.83, and 0.81, and accuracies

(ACCs) of 81.1%, 82.8%, and 78.5% in the training, internal validation, and external test cohorts, respectively (Fig. 2 and Supplementary Table 1). These were significantly higher than the commonly used SUV$_{max}$, which yielded AUCs of 0.62 ($p < 0.001$, Delong test), 0.69 ($p < 0.001$, Delong test), and 0.50 ($p < 0.001$, Delong test), and ACCs of 58.0% ($p < 0.001$, McNemar's test), 72.2% ($p = 0.003$, McNemar's test), and 72.2% ($p < 0.001$, McNemar's test) in the three cohorts, respectively.

When investigating the added value of EGFR-DLS in addition to standard clinical variables (i.e., age, sex, stage, histology, smoking status, and SUV$_{max}$), a clinical signature (CS model) was created by combining sex, histology, and smoking status (all other variables were uninformative), and a combined signature incorporating EGFR-DLS, histology, and smoking status (CMS) were built with multivariable logistic regression analyses in the training cohort. Their quantitative performance shown in Fig. 2 and Supplementary Table 1 indicate that the CMS model had the better performance with AUCs of 0.88, 0.88, and 0.84, ACCs of 82.3%, 82.9%, and 80.0% in the training, internal validation, and external test cohorts, respectively. These were significantly higher than the CS with AUCs of 0.78 ($p < 0.001$, Delong test), 0.78 ($p < 0.001$, Delong test), 0.70 ($p = 0.005$, Delong test), and ACCs of 72.5% ($p < 0.001$, McNemar's test), 72.7% ($p = 0.015$, McNemar's test), and 64.6% ($p = 0.055$, McNemar's test), respectively. However, the difference between the CMS model and the EGFR-DLS by itself was negligible ($p > 0.05$). In addition, through

**Table 2 Demographic and clinical characteristics for TKI-treated and ICI-treated patients.**

| Characteristic | HMU TKI-treated patients (N = 67) | | | P | HLM ICI-treated patients (N = 149) | | | P |
|---|---|---|---|---|---|---|---|---|
| | All | EGFR-DLS | | | All | EGFR-DLS | | |
| | | High (N = 29) | Low (N = 38) | | | High (N = 39) | Low (N = 110) | |
| Age(y) | | | | 0.016* | | | | 0.77 |
| Mean ± SD | 60.31 ± 9.36 | 57.31 ± 8.65 | 62.61 ± 9.33 | | 65.21 ± 13.01 | 65.95 ± 10.36 | 64.95 ± 13.86 | |
| Sex, no. (%) | | | | 0.007* | | | | 0.45 |
| Male | 38 (56.72) | 22 (75.86) | 16 (42.11) | | 60 (40.27) | 18 (46.15) | 42 (38.18) | |
| Female | 29 (43.28) | 7 (24.14) | 22 (57.89) | | 89 (59.73) | 21 (53.85) | 68 (61.82) | |
| TNM stage | | | | 0.26 | | | | 0.15 |
| I | 15 (22.39) | 9 (31.03) | 6 (15.79) | | 0 (0) | 0(0) | 0 (0) | |
| II | 4 (5.97) | 1 (3.45) | 3 (7.89) | | 0 (0) | 0(0) | 0 (0) | |
| III | 15 (22.39) | 4 (13.79) | 11 (28.95) | | 25 (16.78) | 2 (5.13) | 23 (20.91) | |
| IV | 33 (49.25) | 15 (51.72) | 18 (47.37) | | 124 (83.22) | 37 (94.87) | 87 (79.09) | |
| Histology (baseline), no. (%) | | | | 0.016* | | | | 0.048* |
| ADC | 60 (89.55) | 29 (100) | 31 (81.58) | | 100 (67.11) | 21 (53.85) | 79 (71.82) | |
| SCC | 7 (10.45) | 0 (0) | 7 (18.42) | | 49 (32.89) | 18 (46.15) | 31 (28.18) | |
| Smoke, no. (%) | | | | 0.006* | | | | 1.00 |
| Never | 40 (59.7) | 23 (79.31) | 17 (44.74) | | 54 (36.24) | 14 (35.9) | 40 (36.36) | |
| Former | 27 (40.3) | 6 (20.69) | 21 (55.26) | | 95 (63.76) | 25 (64.1) | 70 (63.64) | |
| $SUV_{max}$ | | | | 0.86 | | | | 0.56 |
| Mean ± SD | 9.5 ± 6.97 | 8.93 ± 5.55 | 9.93 ± 7.93 | | 11.67 ± 7.74 | 10.65 ± 6.22 | 12.04 ± 8.21 | |
| PD-L1 status, no. (%) | | | | NA | | | | <.001* |
| ≥50% | NA | | | | 22 (14.77) | 3 (7.69) | 19 (17.27) | |
| 1–49% | | | | | 19 (12.75) | 8 (20.51) | 11 (10.00) | |
| 0% | | | | | 34 (22.82) | 15 (38.46) | 19 (17.27) | |
| Unknown | | | | | 74 (49.66) | 13 (33.33) | 61 (55.45) | |
| Follow up | | | | <0.001* | | | | <0.001* |
| Progression Rate (%) | 34 (50.75) | 7 (24.14) | 27 (71.05) | | 94 (63.09) | 33 (84.62) | 61 (55.45) | |
| PFS, median, month (IQR) | 6.1 (3, 9.36) | 7.5 (3.38,11.15) | 3.75 (2.03,6.10) | | 7.67 (2.76,14.99) | 4.20 (1.77,9.27) | 9.18 (3.67,16.00) | |
| EGFR-deep learning score (EGFR-DLS) | | | | <0.001* | | | | <0.001* |
| Median (IQR) | 0.45 (0.23,0.56) | 0.60 (0.54,0.69) | 0.36 (0.10,0.39) | | 0.29 (0.11,0.53) | 0.64 (0.57,0.72) | 0.21 (0.09,0.35) | |

The comparison of age, EGFR-DLS, and $SUV_{max}$ between two groups was performed using two-sided Wilcoxon sign-rank test, PFS was compared with two-sided log-rank test, and the rest variables were compared with two-sided Fisher's test. ADC is short for adenocarcinoma and SCC is short for squamous cell carcinoma.
IQR interquartile range, NA not available.
*p Value <0.05.

multivariable logistic regression analysis, EGFR-DLS was the only identified significant independent variable in *EGFR* prediction in the validation and test cohorts (Supplementary Table 2).

Multivariate linear regression (adjusted $r^2 = 0.25$, $F = 24.77$, $p < 0.001$) further showed that the EGFR-DLS was independently associated with sex (coefficient = 0.18, $p = 0.007$), histology (coefficient = −0.31, $p < 0.001$), and $SUV_{max}$ (coefficient = −0.14, $p = 0.005$). A total of 25.0% of EGFR-DLS variability could be explained by these three parameters (Supplementary Table 3).

**Distribution and characteristics of EGFR-DLS.** By performing unsupervised hierarchical clustering method on the deeply learned features (i.e., the output of last global average pooling layer, $N = 256$), two patterns were obtained as shown in Fig. 3a. These patterns (I and II) were distinguished by a significantly higher *EGFR* mutation rate ($p < 0.001$), proportion of females ($p < 0.001$), adenocarcinomas ($p < 0.001$), and never smokers ($p < 0.001$) in pattern II for both the training and validation cohorts. Given the limited squamous cell carcinoma rate in the HMU cohort, no significant difference was found between patterns I and II. However, pattern II still had a higher EGFR-mutant rate ($p < 0.001$), female:male ratio ($p = 0.076$), and never smokers ($p = 0.045$) in this cohort. Further, the EGFR-DLS significantly discriminated between the *EGFR*-mutant-type and *EGFR*-wild-type tumors in all three cohorts in both histologies ($p < 0.05$, Fig. 3b).

Figure 3c, d shows representative model inputs and outputs for two patients: one with wild-type *EGFR* and other with mutant *EGFR*. The high-response areas of the filters in the previous layers (the first and second columns of the third line) indicate the self-learned important regions in the subsequent deep learning features generation. After inputting a mutant *EGFR* tumor into the deep learning model, the positive filter (the third column of the third line) generated a strong response, while the negative filter (the fourth column of the third line) was nearly shut down. Similarly, the negative filter was strong activated, and the positive filter was nearly shut down with *EGFR*-wild-type tumor fed to the deep learning model, which reveals the strong classification ability of the deep learning model. When the input (regions of interest) ROIs were enlarged to include more organs and tissues, similar activation maps, positive/negative filters, and predicted EGFR-DLSs were also obtained as shown in Supplementary Fig. 1.

In this work, accurate segmentations were not needed, yet radiologists had to delineate a ROI that contained the tumors and

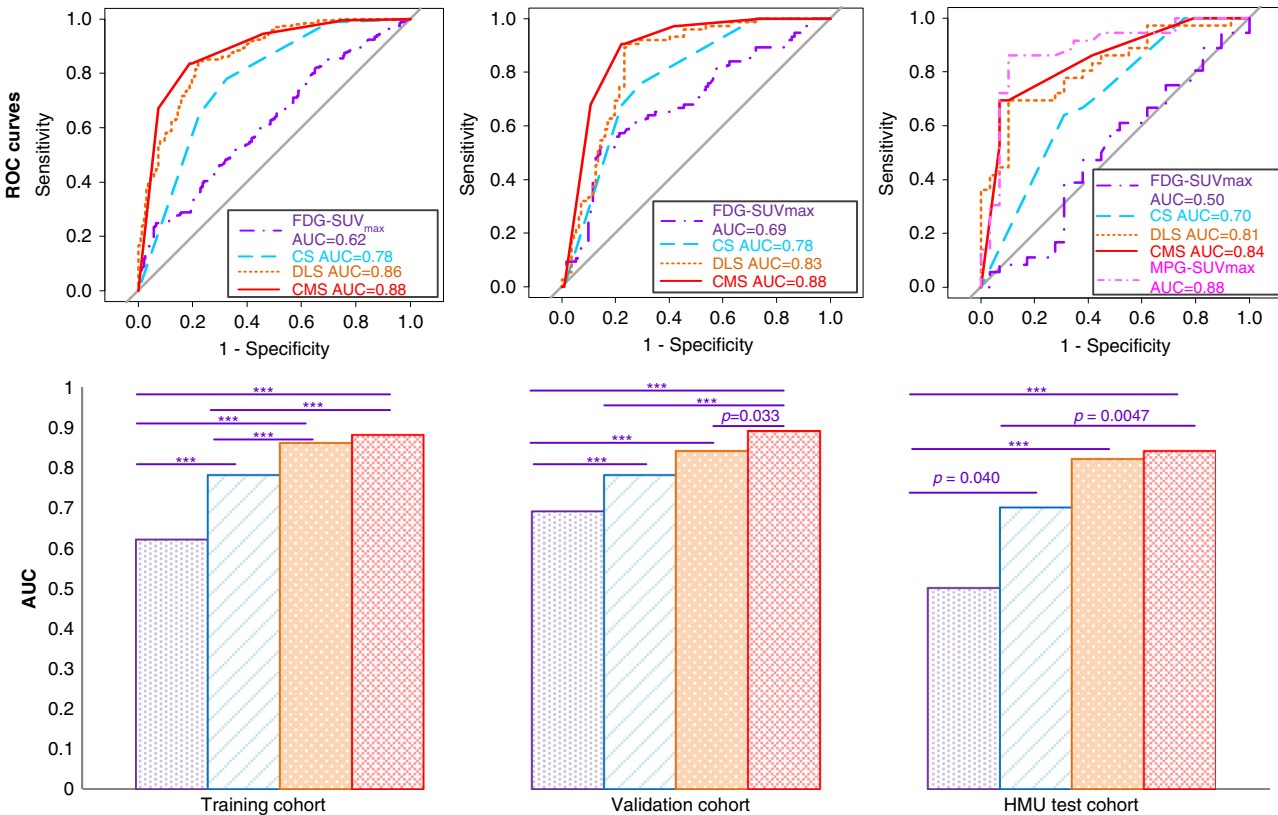

**Fig. 2 Performance of the EGFR-DLS in predicting *EGFR* status across different cohorts.** The top row are the ROC curves of different models in the training, validation, and HMU test cohorts, respectively. The bottom row are the AUC values and the comparison results with Delong test. For statistical comparisons among different models, a two-sided Delong test was used. *** denotes a *p* value <0.001. If *p* value is otherwise, it is noted. Statistics for AUC, sensitivity, specificity, and accuracy for all cohorts are provided in Supplementary Table 1.

some surrounding tissue. To investigate the effect of the minor differences between the different radiologists in selecting the ROIs, the ROIs from a subset of the validation patients (*n* = 73 cases) were generated by all the three radiologists, and three EGFR-DLSs were obtained accordingly. The intraclass correlation coefficient (ICC) of these three EGFR-DLSs was 0.91 (95% confidence interval (CI): 0.87–0.94, *p* < 0.001), and indicating there were no significant differences in AUCs of these three EGFR-DLSs (Supplementary Fig. 2), both of which validate the reproducibility of EGFR-DLS with the input images selected by different radiologists. Further stratified analysis was also performed to investigate the independence of the model on tumor location. For the external HMU test cohort, the EGFR-DLS achieved AUCs of 0.98 (95% CI: 0.93–1.00, *p* = 0.002), 0.80 (95% CI: 0.61–1.00, *p* = 0.020), 0.93 (95% CI: 0.77–1.00, *p* = 0.013), 0.97 (95% CI: 0.88–1.00, *p* = 0.016) in tumors surrounded by air (*n* = 15 cases), tumors surrounded by air and mediastinum (*n* = 23 cases), tumors surrounded by air and chestwall (*n* = 13 cases), and tumors surrounded by air, mediastinum, and chestwall (*n* = 14 cases), respectively. There is no significant difference of the AUC between any two different types (*p* = 0.10–0.82), which suggests this work is independent on tumor location.

**Correlation of EGFR-DLS with histologic findings and MPG imaging.** For the patients with consistent results between *EGFR* status from biopsy and $^{18}$F-MPG imaging (*N* = 64), the EGFR-DLS derived from $^{18}$F-FDG was moderately positively correlated with $^{18}$F-MPG accumulation in tumors measured by $^{18}$F-MPG SUV$_{max}$ (Spearman rho = 0.48, *p* < 0.001, Fig. 4a). Further, the hot-spot regions shown in negative and positive filters (Fig. 3c, d, row 3, columns 3 and 4) also corresponded well with the $^{18}$F-MPG

uptake of the *EGFR*-wild type and mutant type, with a median structural similarity index[21] of 0.66 (interquartile range (IQR): 0.38, 0.77; 0.66 for Fig. 3c and 0.70 for Fig. 3d, respectively).

**Performance of EGFR-DLS to predict EGFR-TKI treatment response.** The distribution of EGFR-DLS in patients with different response is shown in Fig. 4b. In the 31 patients with an objective response (PR/CR) to TKI therapy, the EGFR-DLS was significantly higher (median: 0.53) compared to the 36 patients with PD and SD (median: 0.39; Wilcoxon's *p* = 0.042). Further, if you grouped patients based on the 40 patients with controlled disease (SD/PR/CR), the EGFR-DLS was higher (median: 0.52) though not significant, compared to the 27 patients with PD (median: 0.38; Wilcoxon's *p* = 0.068). The AUCs of binarized EGFR-DLS to identify controlled patients was 0.68 (*p* = 0.012), and 0.67 (*p* = 0.019) to identify the patients with objective response (details shown in Supplementary Table 4). A higher EGFR-DLS (≥0.5) significantly predicted a longer PFS compared to the lower EGFR-DLS (<0.5) group (hazard ratio (HR): 0.24, *p* < 0.001, Fig. 4c and Supplementary Table 5). In addition, the patients with lower EGFR-DLS group and higher EGFR-DLS showed similar PFS compared to the biopsy detected *EGFR*-wild-type patients (*p* = 0.31, log-rank test) and *EGFR*-mutant patients (*p* = 0.91, log-rank test), respectively (Supplementary Fig. 3a). No other clinical characteristics significant in the univariate Cox regression analyses for PFS (Supplementary Table 5).

**Performance of EGFR-DLS in ICI treatment response.** While EGFR-DLS is shown to predict *EGFR* mutation status and response to TKIs, it remains possible that this is merely

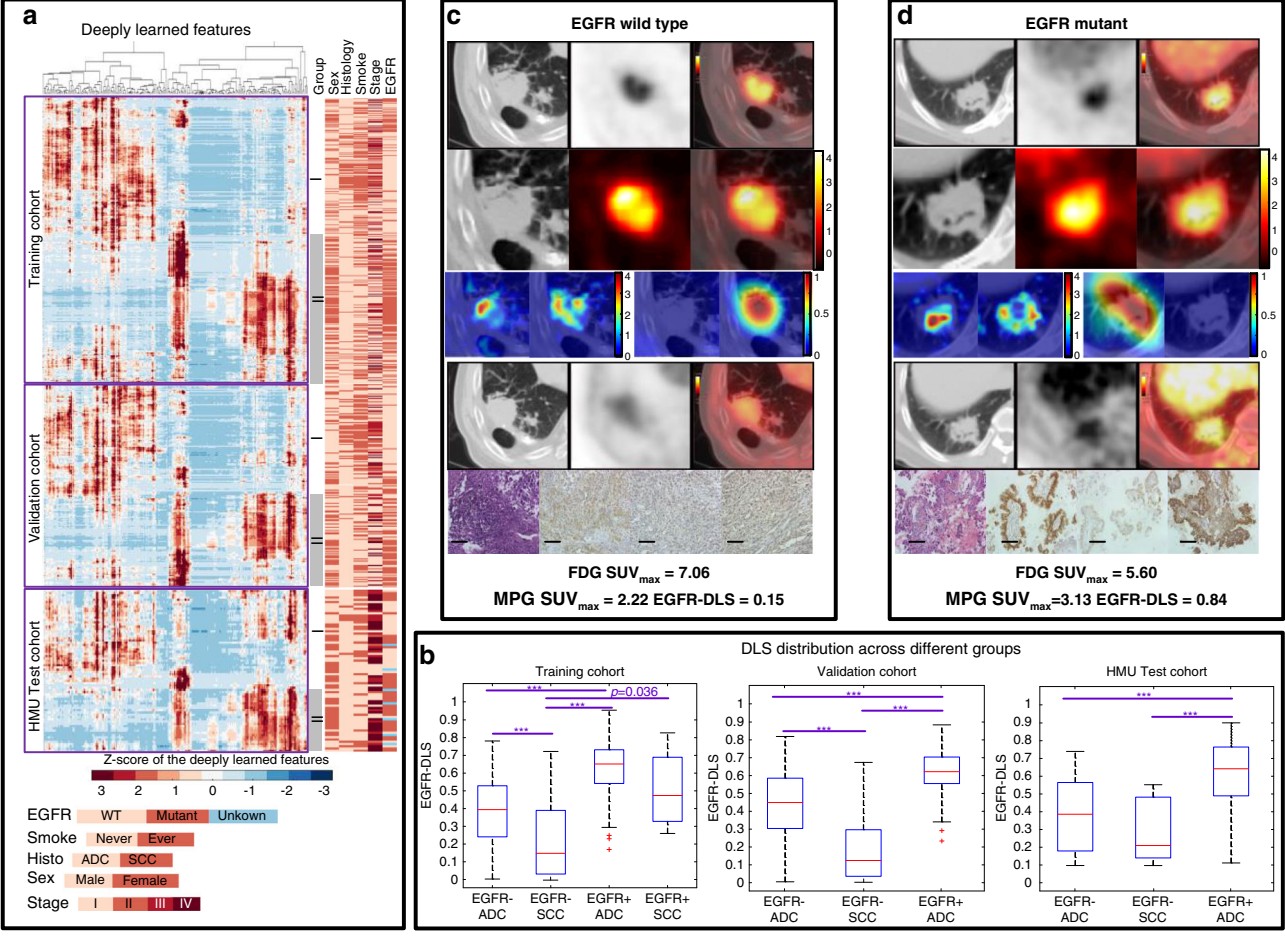

**Fig. 3 The unsupervised clustering of the deep learning features, the distribution of the EGFR-DLS, and two NSCLC adenocarcinoma patients with different *EGFR* mutation status. a** The unsupervised hierarchical clustering of the deep learning features (i.e., the output of global average pooling, $N = 256$) on the vertical axis, which shows a significant association of the deep learning expression patterns with *EGFR* mutation (training: $p < 0.001$, validation: $p < 0.001$, HMU: $p = 0.002$, $\chi^2$ test). There was also significant association of the expression patterns by stage (training: $p < 0.001$, validation: $p < 0.001$, HMU: $p = 0.66$), smoke status (training: $p < 0.001$, validation: $p < 0.001$, HMU: $p = 0.045$), histology (training: $p < 0.001$, validation: $p < 0.001$, HMU: $p = 1.00$), and sex (training: $p < 0.001$, validation: $p < 0.001$, HMU: $p = 0.076$). **b** The EGFR-DLS distribution across different subgroups divided by *EGFR* mutation status and histology type. Significant difference of EGFR-DLS was found between adenocarcinoma (ADC) and squamous cell carcinoma (SCC) for *EGFR*-wild-type patients (training: $p < 0.001$, validation: $p < 0.001$, HMU: $p = 0.24$). In the box plots, the central line represents the median, the bounds of box the first and third quartiles, and the whiskers are the interquartile range. For statistical comparisons among different groups, a two-sided Wilcoxon signed-rank test was used. For the validation cohort, $n = 80$, 32, and 75 for *EGFR*− ADC, *EGFR*− SCC, and *EGFR*+ ADC groups, respectively. For the HMU test cohort, $n = 22$, 7, and 36 for *EGFR*− ADC, *EGFR*− SCC, and *EGFR*+ ADC groups, respectively. Note: ***means $p$ value <0.001. If $p$ value is otherwise it is so noted. **c, d** The patients with wild-type *EGFR* and *EGFR* L858 mutant, respectively. The first lines are the CT, PET, and fusion images of 18F-FDG PET/CT imaging, the second lines are the input ROIs. For the third line, columns 1 and 2 show two of the activation maps of the fourth ResBlock, columns 3 and 4 show the negative filter and positive filter. The fourth lines are the CT, PET, and fusion images of 18F-MPG PET/CT imaging. The last lines show hematoxylin and eosin (H&E) staining, the immunohistochemistry for total-*EGFR*, phospho-*EGFR*, and L858-specific *EGFR* at X20 magnification demonstrating *EGFR* mutation status. Scale bar, 200 μm. Immunohistochemistry scoring was performed on at least two independent biological replicates (slides) per patient.

prognostic and that all patients with elevated EGFR-DLS will respond well, regardless of therapy. To test this, we examined the relationship between the EGFR-DLS and PD-L1 status, and response to ICIs. For the patients with known PD-L1 expression, a weak but significant inverse correlation was observed between the PD-L1 status and EGFR-DLS with Spearman's rho of −0.24 ($p < 0.001$), −0.26 ($p = 0.006$), and −0.26 ($p = 0.024$) for the training, validation, and HLM ICI-treated sub-cohorts, respectively (Supplementary Fig. 4).

Among the ICI-treated patients, 67.27% of the patients with low EGFR-DLS experienced DCB, and this rate was significantly lower (33.33%) for patients with high EGFR-DLS ($p = 0.004$). Notably, 33.33% of patients with high EGFR-DLS responded with

hyperprogression, which was significantly higher than patients with a low EGFR-DLS, who had a rate of 16.36% ($p = 0.037$). Specifically, for the 41 patients with positive PD-L1 status (tumor proportion score (TPS) ≥ 1%), the patients with high EGFR-DLS had a low DCB rate of 54.54% and a high hyperprogression rate of 18.18% vs 76.67% and 6.67% in the patients with low EGFR-DLS. Similar results were obtained for the 34 patients with negative PD-L1 status (TPS = 0%). Those with a high EGFR-DLS had a lower DCB rate of 20.00% and a higher hyperprogression rate of 60.00%, compared to 57.89 and 36.84% in the low EGFR-DLS patients (Supplementary Table 6).

The PFS was significantly longer among ICI-treated patients with low EGFR-DLS compared to those with a high EGFR-DLS

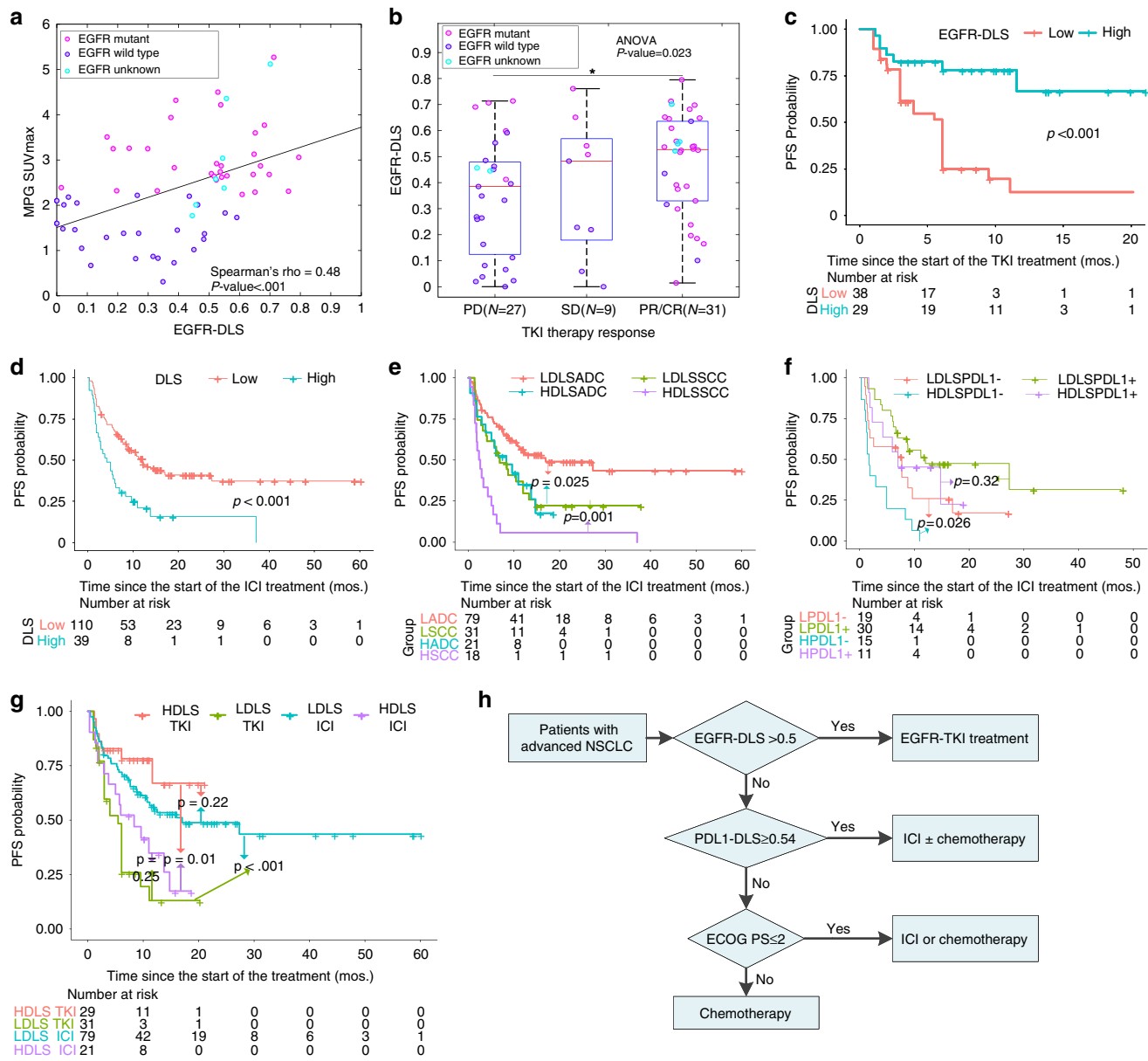

**Fig. 4 Prognostic value of the EGFR-DLS in the different cohorts and in guiding treatment. a–c** The prognostic value of the EGFR-DLS in the TKI-treated cohort. **a** The correlation between the EGFR-DLS and the SUV$_{max}$ of the 18F-MPG PET/CT imaging. $p$ Value indicates two-sided Spearman rank-correlation test. **b** The objective response to TKI treatment relative to the EGFR-DLS. $n = 27$, 9, and 31 for PD, SD, and PR/CR groups, respectively. In the box plot, the central line represents the median, the bounds of box first and third quartiles, and the whiskers are the interquartile range. $p$ Value shows two-sided ANOVA. * denotes a $p$ value <0.05. **c** The progression survival of patients relative the EGFR-DLS (cutoff: 0.5). **d–f** Prognostic value of the EGFR-DLS in the ICI-treated cohorts. **d** The progression survival of patients relative the EGFR-DLS. **e** The progression survival of patients relative the EGFR-DLS and histology type (ADC adenocarcinoma, SCC squamous cell carcinoma). **f** Progression-free survival of patients relative the EGFR-DLS and PD-L1 status (EGFR-DLS cutoff: 0.5). HDLS high EGFR-DLS, LDLS low EGFR-DLS, PD-L1− PD-L1 negative (i.e., the tumor proportion score (TPS) < 1%), PD-L1+ PD-L1 positive (i.e., the tumor proportion score (TPS) ≥ 1%). The LPDL1− patients with low EGFR-DLS and negative PD-L1 status, LPDL1+ patients with low EGFR-DLS and positive PD-L1 status, HPDL1− patients with high EGFR-DLS and negative PD-L1 status, HPDL1+ patients with high-EGFR-DLS and positive PD-L1 status. **g** The progression survival of patients relative the EGFR-DLS and different treatment using the combined TKI-treated and ICI-treated cohorts with adenocarcinoma (EGFR-DLS cutoff: 0.5). HDLS high EGFR-DLS, LDLS low EGFR-DLS. Comparisons of the above progression survival curves were performed with a two-sided log-rank test. **h** Proposed alternative guideline to use EGFR-DLS, PDL1_DLS, and ECOG PS score for decision support for NSCLC patients. ECOG PS Eastern Cooperative Oncology Group performance status.

(12.00 vs 4.20 months; HR: 2.33, $p < 0.001$, as shown in Fig. 4d and Table 2). Stratified analyses by histology and PD-L1 status were performed to investigate the ability of EGFR-DLS to predict outcomes in these subgroups, considering their intimate association with PFS (Supplementary Table 5). The PFS of low EGFR-DLS group was longer than the low EGFR-DLS group in both

ADC and SCC subgroups (Fig. 4e and Supplementary Table 7). The results of the stratified analysis based on PD-L1 status (Fig. 4f and Supplementary Table 8) showed that high EGFR-DLS was still significantly associated with poor outcomes among patients with negative PD-L1 status, i.e., patients with low EGFR-DLS and positive PD-L1 status had the longest PFS, and this was observed

in both histologies (Supplementary Fig. 5, and Supplementary Tables 9 and 10).

**Potential value in guiding treatment**. According to NCCN Guideline Version 2.2020 for treatment of NSCLC[22] (Supplementary Fig. 6), *EGFR* mutation and PD-L1 status through invasive biopsy are two important biomarkers in treatment planning. As an *EGFR* mutation predictor, the value of EGFR-DLS in guiding treatment plan was investigated by analyzing the PFS of combined TKI-treated patients and ICI-treated patients. Since histology is a significant predictor in ICI treatment, and most (89.55%) of the TKI-treated patients were adenocarcinoma, only patients with adenocarcinoma were analyzed in the current study. Through Kaplan–Meier (K–M) analysis (Fig. 4g), for patients with high EGFR-DLS, the PFS of TKI-treated patients was significantly longer than ICI-treated patients ($p = 0.01$), while for patients with low EGFR-DLS, ICIs treatments resulted in a significantly longer PFS ($p < 0.001$). Further, there were no significant differences in PFS between TKI-treated high EGFR-DLS patients and ICI-treated low EGFR-DLS patients.

In addition to the current EGFR-DLS, we have also developed an $^{18}$F-FDG PET/CT-based deep learning score predictor of PD-L1 status (PDL1_DLS), which showed similar prognostic value compared to the IHC-detected PD-L1 status on which it was tested, as shown in Supplementary Fig. 3b and applied it herein[23]. For the patients with high EGFR-DLS (Supplementary Fig. 7a), TKI treatment would improve the PFS significantly compared to ICI in patients with a low-PDL1_DLS ($p = 0.013$). Though there were no significant differences ($p = 0.52$) in the PFS between the two treatments for patients with a high-PDL1_DLS, the TKI-treated patients had an insignificantly higher DCB rate of 80.00% compared to 50.00% for the ICI-treated patients ($p = 0.57$, Fisher's test). Therefore, TKI should be performed on patients with high EGFR-DLS regardless of PDL1_DLS. For the patients with low EGFR-DLS (Supplementary Fig. 7b), patients with high-PDL1_DLS received ICI treatment had significant longer PFS compared to TKI treatment ($p < 0.001$). There is not significant different PFS between two treatment ($p = 0.54$) for the low-PDL1_DLS patients, but ICI treatment could lead to a significant higher 1-year PFS rate (34.29% vs 6.25%, $p = 0.041$, Fisher's test). Therefore, ICI should be performed on patients with low EGFR-DLS and high-PDL1_DLS.

Consequently, an alternative noninvasive guideline (Fig. 4h) could be used in guiding treatment for NSCLC.

## Discussion

Accurate and rapid quantification of *EGFR* mutation status is critical in identifying of lung cancer patients suitable for EGFR-TKI treatment, and provides a potential possibility for guiding ICI immunotherapy. However, the dynamic change in proportion of cells expressing *EGFR* mutation and the invasive tissue-based nature limit the utility of *EGFR* testing compared to image-based assays. Thus, there is a need for a noninvasive, accurate, and reproducible method arises to assess *EGFR* mutation status. In this study, a deeply learning model using PET/CT images was developed to predict *EGFR* mutation status with AUCs of 0.86, 0.83, and 0.81 in the training, validation, and independent test cohorts. This model generates a deeply learned score, EGFR-DLS, whose utility was further validated by identifying patients most likely to benefit by TKI and ICI treatments.

Prior studies have demonstrated the utility of radiomics as an noninvasive approach to predict *EGFR* mutation[20,24]. Liu et al.[24] utilized five CT radiomic features combined with clinical covariates from 298 patients to predict *EGFR* mutational status and found an AUC of 0.71. Wang et al.[20] used transfer learning to

develop and validate a deeply learned predictor based on CT imaging for *EGFR* status with AUC of 0.81. Yip et al.[17] identified the most relevant PET radiomics features for *EGFR* mutation status, with AUC of 0.67 from 387 patients from single institution, and Zhang et al. combined five PET and five CT radiomics features and achieved AUCs of 0.79–0.85 with 248 patients from single institution[18]. In contrast, our analysis yielded among the highest AUCs in the aforementioned studies, but had many advantages including trained and validated with multiple cohorts from four institutions without using accurate tumor segmentations, increasing its generalizability. Further, the clinical utility of the EGFR-DLS related to patient outcomes of TKI and ICI treatments.

Since the uptake of $^{18}$F-MPG is highly correlated with *EGFR* mutation, the generated EGFR-DLS was qualitatively compared to the $^{18}$F-MPG uptake maps. As presented in Fig. 3c, d, the hot-spot regions in negative/positive filter to generate EGFR-DLS corresponded well with the $^{18}$F-MPG uptake regions with high SSIM and the EGFR-DLS was significantly associated with the SUV$_{max}$ of $^{18}$F-MPG, indicating the underlying biological meaning of EGFR-DLS. Furthermore, from the unsupervised clustering of the deeply learned features (Fig. 3a), different histology subtypes have different expression patterns in *EGFR* negative patients, which means the histology type is not requisite when applying the *EGFR* prediction model as presented in Wang et al.'s[20].

We also observed that hyper image constructed with different modalities could significantly improve the accuracy of *EGFR* mutation modeling. By training a similar network only using PET and CT images, the resulting EGFR-DLSs achieved AUCs of 0.76 (95% CI: 0.72, 0.81) and 0.80 (95% CI: 0.76, 0.84) in the training cohort, 0.74 (95% CI: 0.67, 0.81) and 0.75 (95% CI: 0.67, 0.81) in the validation cohort, respectively, which was significantly worse ($p < 0.05$) than those generated using the hyper-images. The similar network with input of PET–CT fused image achieved a lower though not significant AUCs of 0.85 (95% CI: 0.81, 0.88) and 0.79 (95% CI: 0.73, 0.86) in the training ($p = 0.19$) and validation ($p = 0.13$) cohort, respectively. This may be attributed to the important regions used for the accurate prediction of *EGFR* mutation could be better and easier localized by utilizing both metabolic and anatomical information, as reflected by PET and CT images, respectively.

A weak but significant inverse correlation ($-0.26$ to $-0.24$) was observed between the PD-L1 status and the EGFR-DLS. Further, NSCLC harboring *EGFR* mutations were associated with shorter PFS in response to ICI treatment, which is consistent with Kato et al.[25] and Gainor et al.[11], respectively. This could be responsible for the observed poor response to anti-PD-1 treatment among *EGFR*-mutant tumors which are associated with the low rates of PD-L1 expression and CD8+ TILs in *EGFR*-mutant tumors[26]. Importantly, there is addition insight provided by combining the two signatures. As such, we were able to identify a cohort with low EGFR-DLS and low PDL1_DLS, suggesting they may not be responsive to either TKI or ICI (Fig. 4h).

We acknowledge some limitations. First, *EGFR* mutation was usually obtained at the diagnosis of lung cancer, rather than at the initiation of immunotherapy. Second, the patient cohorts were heterogeneous in terms of clinical characteristics and PET/CT image acquisition. However, this can be viewed as a strength, as this heterogeneity decreases the possibility of overfitting to a particular subset of tumors or imaging parameters, and thus will result in a model that is more robust and transportable. Third, only 75 of patients have PD-L1 status in the ICI treatment cohorts, so the complementary information of EGFR-DLS in guiding immunotherapy needed to be validated on a larger cohort with PD-L1 status. Fifth, though 25.0% of EGFR-DLS variability

could be explained by the amalgamation of some standard clinical variables, EGFR-DLS could reflect more information and achieve significant higher performance in predicting *EGFR* mutation status in an easier way, with the more commonly used PET/CT images. Sixth, the hidden colliders like sex, ethnicity, and histology may introduce the selection bias in the current study. Though CNN model with causal inference incorporated provided a good way to reduce this bias[27], not all the patients have the information of these colliders and clinical outcome at the same time. For example, the HLM cohort doesn't have the *EGFR* mutation status, while the SPH and HBMU cohorts don't have the clinical outcome of TKI treatment of ICI treatment. Therefore, this method will be left for future work. In addition, the satisfied results of the test cohorts with different demographic characteristics (e.g., different ethnicities, different histology) further validated that the model was less affected by the hidden colliders. Seventh, given this model is trained mainly for the tumor with 10–20 mm of tumor peripheral region included, the model could not be used for the ROIs without tumor included, and the prediction ability will be decreased with the input of ROI including more organs and tissues. A more intelligent model to solve this problem will be left for our future work. Lastly, this work is based on PET/CT imaging, which is not widely available in many parts of the world. Therefore, this model may be limited to the developed countries and to large urban centers in the developing countries.

In conclusion, an effective and stable deep learning model was identified and may serve as a predictive biomarker to identify NSCLC patients sensitive to EGFR-TKI treatment and to identify patients most likely to benefit from ICI treatment. Due to the advantage of routine acquisition and not subject to sampling bias per se of [18]F-PET/CT images, we prudently propose that this model as a future clinical decision support tool for different treatments pending in larger and prospective trials.

## Methods

**Study population**. In this multi-institutional study, five retrospective cohorts of patients were accrued from four institutions: the SPH, Shanghai, China, the HBMU, Hebei, China, the HMU, Harbin, China, and HLM, Tampa, Florida. Patient cohorts from SPH and HBMU were divided into a training ($n = 429$) and validation cohort ($n = 187$) randomly with a ratio of 70/30 to train, and validate the deep learning model to predict *EGFR* mutation. An EGFR-TKI-treated cohort with *EGFR* status generated in a prospective [18]F-MPG study (ClinicalTrials.gov: NCT02717221 (ref. [15])) at HMU was used as an external test cohort to test this model. Data from cohorts was rigorously kept separate. Then, this EGFR-TKI-treated cohort and an ICI-treated cohort from HLM were used to investigate and validate the association of the generated EGFR-DLS and clinical characteristics on the clinical outcomes of different treatment. Detail of the inclusion criteria are provided in Fig. 1 and Supplementary Methods.

The prognosis values of DLS for EGFR-TKI treatment were investigated through the comparison with target [18]F-MPG molecular imaging, therapy response assessed by CT imaging following standard response criteria: CR, PR, SD, and PD using Response Evaluation Criteria in Solid Tumors (RECIST1.1)[28], as well as PFS.

Hyperprogression (i.e., TTF < 2 months), DCB (PFS ≥ 6 months), and PFS were chosen to investigate the association of the EGFR-DLS and clinical characteristics with the clinical outcome in ICI-treated cohorts. The index date was date of initiation of immunotherapy.

The study was approved by the Institutional Review Boards at the SPH, HBMU, HMU, and University of South Florida, and was conducted in accordance with ethical standards of the 1964 Helsinki Declaration and its later amendments. The requirement for informed consent was waived, as no PHI is reported.

**[18]F-FDG PET/CT Imaging and [18]F-MPG PET/CT imaging**. All patients involved in this study had [18]F-FDG PET/CT imaging. Image acquisition parameters for each cohort are presented in Supplementary Table 11. Since uptake of EGFR-TKI PD153035 based on [18]F-MPG is highly correlated with *EGFR* mutation status[15,29,30], [18]F-MPG PET/CT imaging (Discovery 790 Elite; GE Healthcare) was also performed on HMU cohort. Scanning was initiated 1 h after administration of ~259 MBq of [18]F-MPG. Whole-body CT scans were firstly acquired for attenuation correction by using a low-dose protocol (40 mA, 120 keV), and PET data were subsequently acquired in 3D mode. The anisotropic resolutions for CT and PET images were $0.98 \times 0.98 \times 3.75$ mm$^3$ and $3.65 \times 3.65 \times 3.27$ mm$^3$, respectively[15].

All PET images were converted into SUV units by normalizing the activity concentration to the dosage of [18]F-FDG injected and the patient body weight after decay correction, and all CT images were converted into lung window.

**Tumor *EGFR* and PD-L1 analysis**. All patients in this study underwent surgical resection or biopsy of the primary tumor. The portion of the tumor specimen was carefully examined, and the portion with more malignant cells, less differentiated cells, and less hemorrhage was subjected to histopathological confirmation. The *EGFR* mutation status was determined by ARMS PCR method or gene sequencing. The tumor was identified as *EGFR*-mutant type if any exon mutation was detected; otherwise was regarded as *EGFR*-wild type.

PD-L1 immunohistochemistry was available on 454 patients (training cohort: 267, validation cohort: 112, and HLM ICI-treated cohort: 75), using pharmDx PD-L1 (28-8) rabbit monoclonal antibody and PD-L1 22C3 mouse monoclonal antibody. The PD-L1 expression was presented as a TPS of 0, 1–49, and ≥50%, which is the percentage of viable tumor cells showing membrane PD-L1 staining relative to all viable tumor cells. And PD-L1 positivity was defined as ≥1% of TPS[7].

**Development of the deep learning model**. The *EGFR* mutation status prediction 2D small-residual-convolutional-network (SResCNN) model is presented in Supplementary Fig. 8. The ROIs of the PET and CT images were first selected by experienced nuclear medicine radiologists (L.J., J.Y.Z., and Y.S.) after registration using ITK-SNAP 3.6.0 (ref. [31]) on the condition that entire tumor and at least 10 mm of its peripheral region were included, and were then resized to $64 \times 64$ pixels by spline interpolation and constructed a three-channel hyper image together with their fusion image (alpha-blending fusion[32], $\alpha = 1$; Pipeline is shown in Supplementary Fig. 9). To reduce the effect of the difference between the central slice and peripheral slices, only ROIs that contained measurable tumor tissue were regarded as valid ROIs, and fed into the SResCNN model to update the parameters of the SResCNN model with backward propagation. The *EGFR* mutation status (positive or negative) was encoded to one-hot and used as the label. The output of the network, i.e., the deep learning score (EGFR-DLS), was used as the classification result to represent the *EGFR* mutation positivity probability.

*EGFR* mutation positivity probability at the patient level was obtained by averaging the EGFR-DLSs of the valid slices that included tumor tissue. To reduce overfitting, augmentation including width/height-shift, horizontal/vertical-flip, rotation, and zoom for the 13,583 training hyper-images were used, and the model with the best performance on the validation dataset was selected. Details are shown in Supplementary Methods. The model implemented with Keras toolkit and Python 3.5 (available at https://github.com/lungproject/lungegfr) was further performed on the HMU and HLM cohorts to obtain the EGFR-DLS based on the trained model. ROIs of 73 patients within the validation cohorts were selected by all the three radiologists to validate the reproducibility of EGFR-DLS.

**Visualization of the SResCNN model**. Intermediate activation layers were visualized to see how the network carries the information from input to output[33], and the Gradient-weighted Class Activation Mapping was used to localize the important regions in the input images for predicting the target concept (*EGFR* positive or *EGFR* negative), by using the gradient information of target concept flowing into the last convolutional layer of the SResCNN model, and the reconstructed maps were named as the positive and negative filters[34]. In addition, the deeply learned features (i.e., the output of last global average pooling layer, $N = 256$) were clustered based on the similarities and dissimilarities with unsupervised hierarchical clustering using MATLAB, which was presented by heatmap to show the distinguishable expression pattern among different patients in the training, validation, and external HMU test cohorts, respectively. In order to investigate the correlation between the different patterns and the *EGFR* mutation status (positive or negative), two clusters were chosen to be presented.

**Statistical analysis**. The Wilcoxon signed-rank test and Fisher's exact test were used to test for differences for continuous variables and categorical variables, respectively. One-way ANOVA followed by the Scheffe post hoc test was performed for comparisons involving more than two categories.

The inter-rater agreement of EGFR-DLS estimations were calculated by ICC among the different EGFR-DLSs obtained from the different delineation of the three radiologists. The AUC, ACC, specificity (SPEC), and sensitivity (SEN) with cutoff of 0.5 and the 95% CI by the Delong method[12] were used to assess the ability of EGFR-DLS in discriminating *EGFR*-mutant and *EGFR*-wild type. The median value of the EGFR-DLS from the training cohorts was used as the cutoff. Performance of the EGFR-DLS was likewise compared with other published clinical characteristics, including smoking status, sex, histologic type[35], and PET image-based SUV$_{max}$[36,37] with Delong test. To investigate a potential relation between EGFR-DLS and these indices, stepwise multiple linear regression tests were conducted. Spearman's correlation was used to investigate the relation between EGFR-DLS and PD-L1 status or [18]F-MPG SUV$_{max}$.

K–M survival curves method and Cox proportional hazards model were used to analyze PFS. To rigorously assess the quality of the study design, the radiomic quality score was calculated[38] (Supplementary Methods). Two-sided $p$ values of <0.05 were regarded as statistically significant, and all analyses were conducted

with R 3.5.1 (R Foundation for Statistical Computing, Vienna, Austria) and MATLAB R2019a (Natick, MA).

**Reporting summary**. Further information on research design is available in the Nature Research Reporting Summary linked to this article.

## Data availability

The excel files containing raw data included in the main figures and tables can be found in the Source data file in the article. The PET/CT imaging data and clinical information are not publicly available for patient privacy purposes, but are available from the corresponding authors upon reasonable request (R.J.G. and M.B.S.). The remaining data are available within the article, Supplementary information or available from the authors upon request. Source data are provided with this paper.

## Code availability

The models and the code used to test and evaluate the model is available on GitHub (https://github.com/lungproject/lungegfr)

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

## Acknowledgements

This paper is supported by U.S. Public Health Service research grant U01 CA143062, R01 CA190105, the National Natural Science Foundation of China (81971645, 81627901, and 81471724), the Tou-Yan Innovation Team Program of the Heilongjiang Province (2019-15), Natural Science Foundation of Heilongjiang Province (Grant No.JQ2020H002), National Basic Research Program of China (2015CB931800), and the Key Laboratory of Molecular Imaging Foundation (College of Heilongjiang Province).

## Author contributions

The author meet criteria for authorship as recommended by the International Committee of Medical Journal Editors. W.M., J.T., M.B.S., and R.J.G. contributed to the conception and design of the work; W.M. designed the model and the computational framework and analyzed the data; L.J., J.Z., I.T., J.E.G., C.G., Y.S., X.Z., and X.S. collected the image and clinical data; L.J., J.Z, and Y.S. read and interpreted the images; X.Z., X.S, M.B.S., and R.J.G. supervised the study; M.B.S., and R.J.G. revised the work critical for important intellectual content. All authors contributed to the production of the final manuscript.

## Competing interests

R.J.G. declared a potential conflict with HealthMyne, Inc (Investor (major), Board of Advisors (uncompensated)). J.E.G. reports receiving commercial research grants from AstraZeneca, Merck, Array, Epic Sciences, Genentech, Bristol-Myers Squibb, BI, Tro-vagene, and Novartis, and is a consultant/advisory board member for AstraZeneca, Janssen, Genentech, Eli Lilly, Celgene, and Takeda, and other remuneration from Gen-entech, AstraZeneca, Merck, and Lilly/Genenech. The remaining authors declare no competing interests.
