## [Peer Review File · Nature Communications]

REVIEWER COMMENTS

Reviewer #1 (Remarks to the Author):

It is not clear that the EGFR-DLS model is really guiding treatment, in essence what the authors have done is developed a radiomic biomarker that is predictive of EGFR somatic mutation status, and shown that it is accurate in two cohorts. The subsequent correlation with treatment response is not novel as it is well known that EGFR+ cohorts will benefit from TKI therapy vs.

EGFRnegative/squamous cohorts will benefit from ICI treatment. It is not clear how the EGFR DLS model is actually changing, adding or improving this paradigm?

I strongly suggest to adapt the title to include "using PET/CT imaging", to make it immediately clear what type of data is being used.

In the results section "Distribution and characteristics of EGFR-DLS" it would benefit the reader to concisely provide at least some background of what is meant with pattern I and II. Without reading the methods (which is what most readers will do), it is not possible to understand what these results mean.

What data/cohort was used for the univariate, and multivariate modeling reported in Table S2? This is important as the EGFR-DLS signature was developed likely based on a subset of the data set, and therefore, testing multivariate models combining this signature with EGFR-DLS, ideally this has to be done on a different data set than the one that was used to develop the EGFR-DLS. The same holds for Table S5.

What is the authors interpretation based on the relationship between EGFR-DLS and PD-L1 status? In essence the imaging predictor, EGFR-DLS is a model for EGFR mutation status, so there model is in essence suggesting mutual exclusivity between EGFR mutation status & PDL1 expression, rather than that the EGFR-DLS is predictive of treatment response of ICIs.

Figure 4f needs more clear annotation of the PDL1 status in this figure and in the caption.

Table S8 shows that within the PDL1 negative group, PFS is much lower in the EGFR-DLS high group vs. the EGFR-DLS low group. However, based on the way that the EGFR-DLS model was developed this only suggests that these patients are not receiving the right treatment, as in essence these are predicted to be EGFRmutants and thus should get TKI treatment. Or are the authors suggesting something different here?

To be able to make the point that the EGFR-DLS and the PDL1-DLS (in essence the imaging versions of these molecular biomarkers), are useful in treatment guidance, the performance has to be compared in terms of treatment response with the actual molecular biomarkers. It is not clear if this is anywhere in the manuscript?

A case can be made for non-invasive biomarkers of molecular features, however recent reports show that in the case of driver mutations in EGFR, MET, BRAF & TP53, all these mutations are clonal. Therefore, the heterogeneity argument cannot be made here. See Jamal-Hanjani et al. NEJM 2017.

It is not clear what methods were evaluated in terms of the "accuracy of the segmentations" as virtually no results are reported related to this topic? E.g. this results paragraph:

"For the complete overread study (n=73 cases), high ICCs of 0.91 (95%CI:0.87- 0.94, p<.001) were obtained in three ROIs per cases, and there were no significant differences in AUCs of the EGFR-DLSs obtained with different radiologists' delineations (Supplemental Figure S1)." Appears out of nowhere, the methods section does not detail the segmentation process and how does not suggest any novel way of doing segmentation which would suggest that the model is more robust to

variances in segmentation.

The authors emphasize the “dynamic nature of EGFR” however it is not clear what is meant with that? The authors need to distinguish between EGFR somatic mutation status vs. EGFR expression, as these are two very different things.

It is not clear what the term “residential” is referring to in the name of the CNN “small-residential-convolutional-network”, this seems an error, do the authors actually mean to residual, in the context of ResNet models?

This is unclear “by spline interpolation and constructed a three-channel hyper image together with their fusion image (alpha-blending fusion31, $\alpha=1$)”, the authors need to provide more details how this step was done.

The authors need to explicitly state if the EGFR-DLS model is learning the somatic mutation status positive vs. negative, and not the actual EGFR expression from 18F-MPG-PET imaging to avoid any confusion.

The authors list that 9911 training patches are used? This is unclear, what is a patch, and why are there patches? The model appears to be a 2D model, however, this is nowhere specified, this needs to be made clearer.

Besides Figure 6, not much is mentioned about the model and all its hyperparameters. How was the model architecture decided? What experiments were run to determine it? How where the hyperparameters trained?

The authors should emphasize that an important limitation of their work is the reliance on PET imaging. PET imaging is not available in many parts of the world, and as such this model is going to be limited to academic settings. This also has to be reflected in the title of the manuscript.

Another limitation is that the actual EGFR-DLS is picking up quite some variance from standard clinical variables including sex, histology and SUVmax, and potentially other factors that were not tested? E.g. size, PDL1 status? This is a limitation as in essence it can be seen that the imaging predictor is an amalgamation of clinical and molecular data.

Reviewer #3 (Remarks to the Author):

The paper has a number of strong features like the use of a really large multi-institutional cohort, a fairly well written paper, with some very nice results regarding the classification of EGFR status using the deep learning model and the association with survival using the deep learning model for classifying EGFR status.

However, there are several aspects in which this paper needs improvement.

First and foremost is the methods. There are too few details provided for the methods. For instance, what is the deep learning score? How is this derived? Was it just an average of a bunch of features from a particular layer? How many features were in this layer, if yes?

What is the deep learning model used for training? How was the training done. What was the network that was used? Please present details of training, optimization, learning rates, hyper-

parameter tuning, network architecture etc. Without this, its impossible to understand what these results even mean.

The authors also indicate to two different patterns Pattern I and II, which I presume is based on some form of hierarchical clustering from the Figure 2A. However, these details need to be explained clearly. Again, please explain the clustering algorithm used, how many features were used in the clustering, what layer features were used etc. Were the features from one particular layer or multiple? Which layer features, how this layer was selected and on what dataset was the clustering done etc, how the number of clusters were chosen should be presented.

Results: The strongest results are the classification of the EGFR status and the comparison to the clinical model and the SUV model. My suggestion would be to put this result up front and provide more details of the models trained for the clinical and SUV models to explain the comparison. Also in addition to just the AUCs please provide other metrics like specificity, sensitivity etc to get a better understanding of how well the classifier works.

The next interesting result is the association with survival. It is unclear how the cut off was done to compute the association though. Please explain this in detail. Was this done using median value of the DLS score? Or was ROC analysis done?

Some of the results seem rather speculative. For instance, the result regarding two cancers with hot and cold region filters. From the get go, the radiologists selected the region of interest which very likely did not include a large portion of the background. In that case, its unsurprising that the algorithm would indeed find these regions. A more interesting experiment would be to provide the algorithm with a test set that does not include the ROIs and includes a much larger portion of the image and see what the algorithm does. Also if you report SSIM, its better to report for all the cases instead of just the two examples.

The result regarding the radiologist intra correlation coefficient study is rather confusing. What is the objective here? What is the ICC computed for. Please explain this.

The results regarding the reporting of correlations need to be toned down a bit. There doesn't seem to be any strong correlations. The highest correlation is around 0.48 which is considered moderate. A lot of others even if the Spearman P value is significant are weak to low. Please report them as such. Lack of association (e.g. $P=0.068$) should not be reported as higher.

Reviewer #4 (Remarks to the Author):

The proposed approach is novel and very interesting however there are some points that need to be clarified and discussed. More specifically,

1. The title of the manuscript suggests the development of a treatment decision support system based on retrospective data (observational rather than interventional approach) which raises questions since hidden colliders might kick-in and hamper the clinical value of the proposed DSS. In

particular recent publications (1,2) indicate the necessity to incorporate causal inference in order to estimate the effect of a treatment selection on clinical outcomes when using observational data. Potential hidden colliders like patient fitness, age, ethnicity and others need to be considered in order to reduce selection bias that is inherent to the observational data used in the current study. I would strongly suggest that a relative text should be discussed in the paper, providing arguments why such an approach was not followed.

2. The proposed deep neural network is comprising of almost 1.4 million trainable parameters which is way more than the available samples for training (9911 training patches). The latter raises concerns for overfitting and these needs again to be discussed in a way to provide arguments on how the authors prevented overfitting using such a heavyweight architecture. Also it needs to be very clear that hyper images of the same patient didn't participated in the various data sets simultaneously (training, validation, test).

3. The proposed architecture is 2d, which means that the probabilities for predicting EGFR status is on a slice basis rather than a patient or a tumor basis. How the authors aggregated these probabilities to transform them on a per tumor prediction? Please clarify. The selected architecture also needs some justification, why the authors selected the specific one? Why they didn't consider more suitable approaches including pre-trained networks with transfer learning given the typical $n \ll p$ situation they had?

References

1. van Amsterdam WAC, Verhoeff JJC, de Jong PA, Leiner T, Eijkemans MJC. Eliminating biasing signals in lung cancer images for prognosis predictions with deep learning. NPJ Digit Med. 2019;2:122. Published 2019 Dec 10. doi:10.1038/s41746-019-0194-x
2. Paul W Holland. Statistics and causal inference. Journal of the American statistical Association, 81(396):945–960, 1986.

Response to REVIEWER #1

1. It is not clear that the EGFR-DLS model is really guiding treatment, in essence what the authors have done is developed a radiomic biomarker that is predictive of EGFR somatic mutation status, and shown that it is accurate in two cohorts. The subsequent correlation with treatment response is not novel as it is well known that EGFR+ cohorts will benefit from TKI therapy vs. EGFR negative/squamous cohorts will benefit from ICI treatment. It is not clear how the EGFR DLS model is actually changing, adding or improving this paradigm?

Reply:

Thanks very much for the thoughtful critique.

At present, EGFR status detected by tissues collected through invasive biopsy or ctDNA analysis from plasma is a clinically actionable biomarker for treatment planning according to NCCN Guideline for treatment of Non-Small Cell Lung Cancer. However, the EGFR mutation determined by tissue-based assays has many limitations including *inter alia*: a requirement for invasive biopsies with associated morbidities, the assays are not rapid, and may fail to yield actionable results due to insufficient quantity or quality of the tissue. EGFR mutation determined by plasma based assays have limitations including requiring active shedding of tumor DNA into the blood stream which does not occur in all patients, thus creating a subset of false negative results. Additionally, EGFR mutational status may change during the course of therapy and progression, which may need repeated sampling. The results from our study using the EGFR-DLS provide a complimentary and possibly alternative non-invasive guideline for treatment planning, which could reduce the risk of repeated invasive biopsies.

Having said that, we agree with the reviewer that it is well known that EGFR positive cohorts will benefit from TKI therapy and that EGFR negative cohorts will have better benefit from ICI treatment. However, while a decision could be made on EGFR status alone, we believe that knowledge of both EGFR status and PD-L1 status would make for a stronger decision support paradigm. While PD-L1 status and EGFR mutation status could be determined by IHC and advanced DNA sequencing of the same biopsy, there are the above-described limitations in tissue testing and these could be complimented by EGFR and PD-L1 status determined by the same information-rich PET/CT scan. Non-invasive image biomarkers to distinguish between two treatments at the same time haven't been well investigated.

Therefore, due to the availability of the routine PET/CT images, the non-invasive, easy-to-use characteristics, and real-time turn-around time to results, the deep learning based model provides a complimentary confirmatory data point in the case that high quality IHC are available, and an alternative when the EGFR or PD-L1 status is not available due to the unavailable or insufficient tissue samples.

2. I strongly suggest to adapt the title to include "using PET/CT imaging", to make it immediately clear what type of data is being used.

Reply:

We have revised the title to **"Non-invasive decision support for NSCLC treatment using PET/CT Radiomics"**.

3. In the results section "Distribution and characteristics of EGFR-DLS" it would benefit the reader to concisely provide at least some background of what is meant with pattern I and II. Without reading the methods (which is what most readers will do), it is not possible to understand what these results mean.

Reply:

Thank you for your useful suggestion. Accordingly, we have added concise detail in both the results and methods section as follows.

“Results

Distribution and characteristics of EGFR-DLS

By performing an unsupervised hierarchical clustering on the deeply learned features (i.e., the output of last global average pooling layer, N=256), two patterns were obtained, as shown in Figure 3a. These patterns (I and II) were distinguished by a significantly higher EGFR mutation rate ($p<.001$), proportion of females ($p<.001$), adenocarcinomas ($p<.001$), and never smokers ($p<.001$) in pattern II for both the training and validation cohorts.

Patients and Methods

Visualization of the SResCNN model

Additionally, the deeply learned features (i.e., the output of last global average pooling layer, N=256) were clustered based on the similarities and dissimilarities with unsupervised hierarchical clustering using MATLAB, which was presented by heatmap to show the distinguishable expression pattern among different patients in the training, validation and external HMU test cohorts respectively.”

4. (a) What data/cohort was used for the univariate, and multivariate modeling reported in Table S2? This is important as the EGFR-DLS signature was developed likely based on a subset of the data set, and therefore, testing multivariate models combining this signature with EGFR-DLS, ideally this has to be done on a different data set than the one that was used to develop the EGFR-DLS. (b)The same holds for Table S5.

Reply:

Thank you for your comments and we agree.

(a) Initially, Table S2 showed only the results of the training cohort. In order to investigate the added value of EGFR-DLS to standard clinical variables, a clinical model comprised of standard clinical variables, and a combined model with EGFR-DLS plus clinical variables were trained using multivariable logistic analysis and subsequently tested on independent validation and test cohorts. Additionally, their predictive performance was compared with just the EGFR-DLS in different cohorts based on AUC, accuracy, specificity, and sensitivity. To make this Table clearer, and to further validate the important value of EGFR-DLS in different cohorts, we have now added the results of the multivariable logistic analysis of the validation and test cohorts in Table S2 as follows. All three multivariable analyses (training, validation, and test cohorts) identified the EGFR-DLS as being independently significant. Notably, in the external HMU test cohort, only EGFR-DLS was identified as being significant. Therefore, these results further indicate the important predictive value of the EGFR-DLS compared to the clinical variables. This has now been added to the third paragraph of RESULTS:

“Performance of EGFR-DLS in predicting EGFR mutation status”

...

When investigating the added value of EGFR-DLS in addition to standard clinical variables (age, sex, stage, histology, Smoking status, and SUVmax), a clinical signature (CS model) was created by combining sex, histology, and smoking status (all other variables were uninformative), and a combined signature incorporating EGFR-DLS, histology and smoking status (CMS model) were built with

multivariable logistic regression analyses in the training cohort. Their quantitative performance shown in **Figure 2** and Supplemental **Table S1** indicate that the CMS model had the better performance with AUCs of 0.88, 0.88, and 0.84, ACCs of 82.3%, 82.9% and 80.0% in the training, internal validation, and external test cohorts, respectively. These were significantly higher than the CS with AUCs of 0.78 (p<0.001, Delong test), 0.78 (p<0.001, Delong test), 0.70 (p=0.005, Delong test), and ACCs of 72.5% (p<0.001, McNemar's test), 72.7% (p=0.015, McNemar's test) and 64.6% (p=0.055, McNemar's test), respectively. However, the difference between the CMS model and the EGFR-DLS by itself was negligible (p>0.05).

Additionally, through multivariable logistic regression analysis, EGFR-DLS was the only identified significant independent variable in EGFR prediction in the validation and test cohorts (Supplemental Table S2)."

Table S2. Logistic regression analysis of factors for EGFR prediction

Training cohort	Univariable		Clinical signature		Combined signature	
	Odds Ratio(95% CI)	p	Odds Ratio (95% CI)	p	Odds Ratio (95% CI)	p
Age	0.99 (0.99-1.00)	0.17				
Sex	1.11(0.98-1.26)	0.096	2.44 (1.30-4.59)	0.006		
Stage	0.96 (0.85-1.01)	0.088				
Histology	0.75(0.64-0.87)	<.001	0.093(0.03-0.27)	0.006	0.16 (0.052-0.52)	0.002
Smoke	0.30(0.22-0.42)	<.001	0.45 (0.23-0.85)	0.014	0.24 (0.14-0.42)	<.001
SUVmax	0.97 (0.95-0.99)	0.001				
EGFR-DLS	19.52(11.9-32.02)	<.001			14.53 (8.50-24.86)	<.001
Constant			4.71	0.063	2.71	0.13

Validation cohort	Univariable		Clinical signature		Combined signature	
	Odds Ratio(95% CI)	p	Odds Ratio(95% CI)	p	Odds Ratio (95% CI)	p
Age	1.00 (0.97-1.03)	0.93				
Sex	7.74(3.99-15.03)	<.001	5.07(2.54-10.13)	<.001	5.91(2.54-13.77)	<.001
Stage	0.92 (0.71-1.21)	0.57				
Histology	0.036(0.01-0.27)	.001	0.094(0.012-0.74)	.025		
Smoke	0.16(0.084-0.31)	<.001				
SUVmax	0.88(0.82-0.94)	<.001	0.93(0.87-1.00)	0.055		
EGFR-DLS	30.58(12.55-74.5)	<.001			25.77(10.06-66.0)	<.001
Constant			1.52	0.93	0.007	<.001

HMU test cohort	Univariable		Clinical signature		Combined signature	
	Odds Ratio(95% CI)	p	Odds Ratio (95% CI)	p	Odds Ratio (95% CI)	p
Age	0.98(0.93-1.03)	0.35				
Sex	2.83(1.03-7.80)	0.044	2.83(1.03-7.80)	0.044		
Stage	1.12 (0.74-1.67)	0.60				
Histology	-	-				
Smoke	0.36(0.13-0.99)	.048				
SUVmax	1.01 (0.94-1.08)	0.81				
EGFR-DLS	19.70(4.91-79.05)	<.001			19.70 (4.91-79.05)	<.001
Constant			0.25	0.095	0.42	0.017

Note., For Sex: male was assigned set as the referent group and for histology, adenocarcinoma was set as the referent group

(b) Table S5 now shows the results of the Cox analysis based on the independent external TKI-treated and ICI-treated test cohorts, and we have clarified this in the revised version as follows.

Table S5. Univariable Cox analysis of risk factors for PFS on independent TKI-treated and ICI-treated cohorts

	TKI-patients		ICI-patients	
	Hazard ratio (95% CI)	p	Hazard ratio (95% CI)	p
Age	1.03(0.99-1.08)	0.099	0.99 (0.98-1.01)	0.79
Sex	0.84 (0.43-1.64)	0.61	0.73 (0.48-1.11)	0.14
Stage	1.07 (0.80-1.42)	0.65	1.02 (0.71-1.46)	0.92
Histology(baseline)	1.29 (0.45-3.67)	0.64	2.19 (1.45-3.31)	<.001
Smoke	1.17 (0.59-2.30)	0.66	0.95(0.42-2.15)	0.89
SUVmax	0.95 (0.90-1.02)	0.97	1.00 (0.98-1.03)	0.93
EGFR-DLS	0.24 (0.11-0.57)	<.001	2.33 (1.51-3.60)	<.001
PD-L1 status	NaN		0.39 (0.22-0.68)	0.001

Note., For Sex: male was assigned set as the referent group and for histology, adenocarcinoma was set as the referent group

5. What is the authors' interpretation based on the relationship between EGFR-DLS and PD-L1 status? In essence the imaging predictor, EGFR-DLS is a model for EGFR mutation status, so there model is in essence suggesting mutual exclusivity between EGFR mutation status & PDL1 expression, rather than that the EGFR-DLS is predictive of treatment response of ICIs.

Reply:

Thank you for your thoughtful comments. According to Gainor et al¹¹, EGFR mutant NSCLC patients had significantly lower rates of PD-L1 expression and CD8+ TILs compared to EGFR WT. Notably, CD8⁺ TIL cells are generally thought to be the dominant effector population following ICI treatment, and a lack of effector cells may limit an antitumor immune response, regardless of PD-L1 status. Therefore, the low rates of PD-L1 expression and CD8+ TILs may contribute to the low response rates to ICI treatment among EGFR mutant patients. Given the EGFR-DLS has a high predictive value for EGFR mutational status, the EGFR-DLS alone could be predictive of treatment response of ICIs, as the reviewer states. However, this is a passive diagnosis and we have preferred instead to generate a signature for PD-L1 status. Indeed, we did observe a weak but significant inverse correlation between the two signatures. Consistent with this, as we now show in Fig. 4h, using both signatures we were able to identify a sizeable cohort who had low EGFR-DLS and low PDL1_DLS, suggesting that they may not be response to either TKI or ICI. We have altered this in the discussion accordingly.

“A weak but significant inverse correlation (-0.26~-0.24) was observed between the PD-L1 status and the EGFR-DLS. Further, NSCLC harboring EGFR mutations were associated with shorter PFS in response to ICI treatment, which is consistent with Kato et al.²⁵ and Gainor et al¹¹, respectively. This could be responsible for the observed poor response to anti-PD-1 treatment among EGFR-mutant tumors with low rates of PD-L1 expression and low CD8+ TILs in EGFR-mutant tumors²⁶. Importantly, there is addition insight provided by combining the two signatures. As such, we were able to identify a cohort with low EGFR-DLS and low PDL1_DLS suggesting that they may not be responsive to either TKI or ICI

(Fig. 4h)."

6. Figure 4f needs more clear annotation of the PDL1 status in this figure and in the caption.

Reply:

Thanks for pointing this out. We apologize for the lack of clarity; the caption has been revised to include additional details:

"f) is the progression survival of patients relative the EGFR-DLS and PD-L1 status (EGFR-DLS cutoff:0.5). HDLS means high-EGFR-DLS, LDLS means low-EGFR-DLS, PDL1- means the PD-L1 negative (i.e. the tumor proportion score (TPS) <1%), PDL1+ means the PD-L1 positive (i.e. the tumor proportion score (TPS) ≥1%). The LPDL1- means the patients with low-EGFR-DLS and negative PD-L1 status, the LPDL1+ means the patients with low-EGFR-DLS and positive PD-L1 status, The HPDL1- means the patients with high-EGFR-DLS and negative PD-L1 status, and the HPDL1+ means the patients with high-EGFR-DLS and positive PD-L1 status."

7. Table S8 shows that within the PDL1 negative group, PFS is much lower in the EGFR-DLS high group vs. the EGFR-DLS low group. However, based on the way that the EGFR-DLS model was developed this only suggests that these patients are not receiving the right treatment, as in essence these are predicted to be EGFR mutants and thus should get TKI treatment. Or are the authors suggesting something different here?

Reply:

Thank you for your comments. This is exactly what we would like to say, but we cannot because this was a retrospective study, and thus we could not test this hypothesis. In this study, the ICI-treated patients were curated from different clinical trials. The inclusion criteria for some of these trials did not account for EGFR mutation status.

The design of our study includes three parts: the predictive value of EGFR-DLS in TKI treatment; the predictive value of EGFR-DLS in ICI treatment; and the potential value of EGFR-DLS in guiding treatment together with the PDL1_DLS. Table S8 demonstrates that EGFR-DLS was a significant negative predictive factor for ICI treatment, which corresponded to the second part of the study design. The final alternative non-invasive guideline (i.e., the third part) was obtained according to the further analysis on the TKI-treated and ICI-treated patients. Therefore, we think this is not conflicting or inconsistent.

8. To be able to make the point that the EGFR-DLS and the PDL1_DLS (in essence the imaging versions of these molecular biomarkers), are useful in treatment guidance, the performance has to be compared in terms of treatment response with the actual molecular biomarkers. It is not clear if this is anywhere in the manuscript?

Reply:

At your suggestion, we have performed this analysis. We have added the comparison of progression free survival between the molecular/tissue-based biomarkers (EGFR and PD-L1) and image-based biomarkers (EGFR-DLS and PDL1_DLS) in the revised manuscript. The revised Supplemental Figure S3 (below) shows that the PFS probabilities were indistinguishable between the DLS and the molecular scores. The advance herein is

that these non-invasive radiomic markers are readily obtainable and can be used to predict EGFR and PDL1 status. These have also been clarified in the main text as follows:

“Performance of EGFR-DLS to predict EGFR-TKI-treatment response

Additionally, the patients with lower EGFR-DLS group and higher EGFR-DLS showed similar PFS compared to the biopsy detected EGFR wild type patients ($p=0.31$) and EGFR mutant patients ($p=0.91$), respectively (Supplemental Figure S3a).

Potential value in guiding treatment

In addition to the current EGFR-DLS, we have also developed an ^{18}F -FDG PET/CT based deep learning score predictor of PD-L1 status (PDL1_DLS), which showed similar prognostic value compared to the IHC-detected PD-L1 status on which it was tested, as shown in Supplemental Figure S3b and applied it herein.”

Figure S3. Comparison of progression survival between the tissue-based molecular biomarkers (EGFR and PD-L1) and image-based biomarkers (EGFR-DLS and PDL1_DLS). (a) is the progression survival of TKI-treated patients with EGFR mutation status relative to the EGFR mutation and EGFR-DLS (cutoff:0.5). (b) is the progression survival of ICI-treated patients with PD-L1 status relative to the EGFR-DLS (cutoff:0.5) and PD-L1 status. Note. *P* value was from log rank test.

9. A case can be made for non-invasive biomarkers of molecular features; however recent reports show that in the case of driver mutations in EGFR, MET, BRAF & TP53, all these mutations are clonal. Therefore, the heterogeneity argument cannot be made here. See Jamal-Hanjani et al. NEJM 2017.

Reply:

Indeed, we agree that the results coming out of TRACERx studies are eye-popping and informative. They do seem to indicate that “driver” mutations are generally clonal (i.e. present in all cancer cells). However, in figure 4 of Jamal-Hanjani, it can be seen that EGFR mutations was sub-clonal in 2/13 samples, which could lead to false negatives. Nonetheless, because of the reviewers’ concern, which we are sure others share, we have

removed reference to heterogeneity. The discussion has been revised as follows.

*“Accurate quantification of EGFR mutation status is critical in identifying of lung cancer patients suitable for EGFR-TKI treatment and provides a potential possibility for guiding ICI immunotherapy. However, the dynamic nature of EGFR mutation and **the invasive nature of tissue-based** methods limit the utility of EGFR testing **compared to image-based assays.**”*

10. It is not clear what methods were evaluated in terms of the “accuracy of the segmentations” as virtually no results are reported related to this topic? E.g. this results paragraph:

“For the complete overread study (n=73 cases), high ICCs of 0.91 (95%CI: 0.87- 0.94, p<.001) were obtained in three ROIs per cases, and there were no significant differences in AUCs of the EGFR-DLSs obtained with different radiologists’ delineations (Supplemental Figure S1).” **Appears out of nowhere, the methods section does not detail the segmentation process and how does not suggest any novel way of doing segmentation which would suggest that the model is more robust to variances in segmentation.**

Reply:

Thank you so much for your comments, as we had overlooked this description in Methods.

In this work, rough ROIs were selected by radiologist using ITK software (the analytic Pipeline is shown in Supplemental Figure S9) and accurate segmentations were not needed, which is one advantage of Deep Learning over Conventional Radiomics. Therefore we didn’t evaluate the accuracy of the segmentations. However, because there could be minor differences among different radiologists in selecting the rough ROIs, we had all three radiologists delineate ROIs of 73 patients within the validation cohort to examine its effects on the reproducibility of DLS. The inter-rater agreement of DLS estimations were calculated by intraclass correlation coefficient (ICC) among the three radiologists. We have now clarified this in both Results and Methods as follows.

“Results

Distribution and characteristics of EGFR-DLS

In this work, accurate segmentations were not needed, yet radiologists had to delineate a rough ROI that contained the tumors and some surrounding tissue. In order to investigate the effect of the minor differences between different radiologists in selecting the rough ROIs, the ROIs of a subset of the validation patients (n=73 cases) were generated by all the three radiologists, and three EGFR-DLSs were obtained accordingly. The intraclass correlation coefficient (ICC) of these three EGFR-DLSs was 0.91 (95%CI:0.87-0.94,p<.001), indicating that there were no significant differences in AUCs of these three EGFR-DLSs (Supplemental Figure S2), both of which validate the reproducibility of DLS with the input images selected by different radiologists. “

“Patients and Methods

ROIs of 73 patients within the validation cohorts were selected by all the three radiologists to validate the reproducibility of EGFR-DLS.

The inter-rater agreement of EGFR-DLS estimations were calculated by intraclass correlation coefficient (ICC) among the different EGFR-DLSs obtained from the different delineation of the three radiologists. “

The details of the segmentation process were provided in supplementary S2 and supplementary Figure S9 as follows.

“Preparation of the input images

*Using ITK-SNAP software, the PET and CT images were firstly registered, and then a square or an irregular box that was close to the boundary of the tumor was delineated, by experienced nuclear medicine radiologist. After resampling, dilation, and resize using cubic spline interpolation, the PET region of interest (ROI) and CT ROI were obtained keeping the entire tumor and its peripheral region with the same size (64×64). (Pipeline is shown in **Supplemental Figure S9**). “*

11. The authors emphasize the “dynamic nature of EGFR” however it is not clear what is meant with that? The authors need to distinguish between EGFR somatic mutation status vs. EGFR expression, as these are two very different things.

Reply:

Thank you so much for your suggestion.

EGFR mutational status might change during the course of therapy and subsequent clonal selection for resistant variants. While in the case of EGFR mutations, resistance is commonly associated with point mutations, this is not always the case, such as amplification of Met pathway (e.g. Ricordel et al, Ann Oncol, 2018) that can abrogate the need for EGFR. And in order to make it clearer, we have revised to “dynamic change in proportion of cells expressing EGFR mutation” in the revised version.

Additionally, we have changed “EGFR expression” to “EGFR mutation status” in describing the EGFR-DLS and the ¹⁸F-MPG.

12. It is not clear what the term “residential” is referring to in the name of the CNN “small-residential-convolutional-network” , this seems an error, do the authors actually mean to residual, in the context of ResNet models?

Reply:

We apologize for this error; we have revised this to “small-residual-convolutional-network” in the current version.

13. This is unclear “by spline interpolation and constructed a three-channel hyper image together with their fusion image (alpha-blending fusion31, $\hat{I}\pm=1$)” , the authors need to provide more details how this step was done.

Reply:

Thank you for your suggestion, and we have added the details in the supplementary S2 as follows.

“Supplemental S2: Details of the deep learning model

Preparation of the input images

Using ITK-SNAP software, the PET and CT images were firstly registered, and then a square or an irregular box that was close to the boundary of the tumor was delineated, by experienced nuclear medicine radiologist. After resampling, dilation, and resize using cubic spline interpolation, the PET region of interest (ROI) and CT ROI were obtained keeping the entire tumor and its peripheral region with the same size (64×64). Subsequently, the fusion images were calculated through the α -fusion equation:

$$I_{FUSE} = I_{PETnorm} + I_{CTnorm}$$

where $I_{PETnorm}$ and I_{CTnorm} are the normalized PET and CT pixel-wise image data by z-score normalization. The fusion ROI was further standardized by z-score normalization, and constructed a 3-channel hyper-image together with the normalized PET ROI and normalized CT ROI. This hyper-image was used as the input of the SResCNN model (Pipeline is shown in **Supplemental Figure S9**). Z-score normalization means the ROI image was subtracted by the mean intensity value and divided by the standard deviation of the image intensity, before inputting to the deep learning model, to reduce the effect of different equipment and different reconstruction parameters.”

14. The authors need to explicitly state if the EGFR-DLS model is learning the somatic mutation status positive vs. negative, and not the actual EGFR expression from 18F-MPG-PET imaging to avoid any confusion.

Reply:

Thank you for your suggestion. We have changed all “EGFR expression” to “EGFR mutation status” in describing the EGFR-DLS. Additionally, this has been further clarified in the development of the DLS as follows.

“The EGFR mutation status (positive vs. negative) was encoded to one-hot and used as the label. The output of the network, i.e., the deep learning score (EGFR-DLS), was used as the classification result to represent the EGFR mutation positivity probability.”

15. The authors list that 9911 training patches are used? This is unclear, what is a patch, and why are there patches? The model appears to be a 2D model, however, this is nowhere specified, this needs to be made clearer.

Reply:

We apologize for this error. It should be the ROI of the constructed hyper-images. And the size of the training data should be 13,583, and we have revised in this version as follows:

*“To reduce overfitting, augmentation including width/height-shift, horizontal/vertical-flip, rotation and zoom for the 13,583 training ROI-based **hyper-images** were used”*

We have also defined this as a 2D model in the methods and supplemental S2 as follows.

“The EGFR mutation status prediction 2D small-residual-convolutional-network (SResCNN) model is presented in Supplemental Figure S8.

Structure and training of the SResCNN network

The 2D SResCNN is based on several residual blocks with the 3-channel input images, which is similar to the well-known Resnet18 network with fewer filters.”

16. Besides Figure S6, not much is mentioned about the model and all its hyperparameters. How was the model architecture decided? What experiments were run to determine it? How were the hyperparameters trained?

Reply:

We appreciate your concerns and addressing them will improve the portability and quality of this work. To make the manuscript more concise, we omitted some pertinent details of the deep learning model. Resnet is a popular and powerful architecture for image classification. Given the limited dataset, we constructed the network with fewer layers, similar to Resnet-18 with fewer filters, which was trained on our own workstation. Additionally, given single resolution may not be optimal and depends on the scale of the objects within the image, a multi-resolution CNN model was proposed and proved to have significantly better performance [s1, s2]. Therefore, we incorporated the concept of multi-resolution into this small Resnet structure. By trying different number of filters of each layer, the current structure could give an acceptable result with fewest filters on the validation cohort. In response, we have now added more details about the network and the training process in the **Patients and Methods** and **Supplemental methods S2** as follows.

“Patients and Methods

Development of the deep learning model

*The EGFR mutation status prediction 2D small-residual-convolutional-network (SResCNN) model is presented in **Supplemental Figure S8**. The regions of interest (ROIs) of the PET and CT images were first selected by experienced nuclear medicine radiologists (L.J, JY. Z, and Y.S) after registration using ITK³⁰ on the condition that entire tumor and at least 10 mm of its peripheral region were included, and were then resized to 64x64 pixels by spline interpolation and constructed a three-channel hyper image together with their fusion image (alpha-blending fusion³¹, $\alpha=1$) (Pipeline is shown in **Supplemental Figure S9**). To reduce the effect of the difference between the central slice and peripheral slices, only ROIs that contained measurable tumor tissue were regarded as valid ROIs, and were fed into the SResCNN model to update the parameters of the SResCNN model with backward propagation. The EGFR mutation status (positive/negative) was encoded to one-hot and used as the label. The output of the network, i.e. the deep learning score (EGFR-DLS), was used as the classification result to represent the EGFR mutation positivity probability.*

*EGFR mutation positivity probability at the patient level was obtained by averaging the EGFR-DLSs of the slices that included tumor tissue. To reduce overfitting, augmentation including width/height-shift, horizontal/vertical-flip, rotation and zoom for the 13,583 training hyper-images were used, and the model with the best performance on the validation dataset was selected. **Details are shown in Supplemental S2.***

“Supplemental S2: Details of the deep learning model

Preparation of the input images

Using ITK-SNAP software, the PET and CT images were firstly registered, and then a square or an irregular box that was close to the boundary of the tumor was delineated, by experienced nuclear medicine radiologist. After resampling, dilation, and resize using cubic spline interpolation, the PET region of interest (ROI) and CT ROI were obtained keeping the entire tumor and its peripheral region

with the same size (64×64). Subsequently, the fusion images were calculated through the α -fusion equation:

$$I_{FUSE} = I_{PETnorm} + I_{CTnorm}$$

where $I_{PETnorm}$ and I_{CTnorm} are the normalized PET and CT pixel-wise image data by z-score normalization. The fusion ROI was further standardized by z-score normalization, and constructed a 3-channel hyper-image together with the normalized PET ROI and normalized CT ROI. This hyper-image was used as the input of the SResCNN model (Pipeline is shown in **Supplemental Figure S9**). Z-score normalization means the ROI image was subtracted by the mean intensity value and divided by the standard deviation of the image intensity, before inputting to the deep learning model, to reduce the effect of different equipment and different reconstruction parameters. Because of the big difference of the central slice and peripheral slices, only the slices with the area larger than the 30% of the maximum area of this patient were regarded as valid input images and were used as the input of the deep learning model. The area here means the area of the smallest square including the selected region (Supplemental Fig S9c). Finally, 13,583 training hyper-images were generated for training.

Structure and training of the small-residual-convolutional-network (SResCNN)

The 2D SResCNN is based on several residual blocks with 3-channel input images, which is similar to the well-known Resnet18 network with fewer filters. Given single resolution may not be optimal and depends on the scale of the objects within the image, multi-resolution CNN model was proposed and proved to have significantly better performance [s1, s2]. Therefore, the concept of multi-resolution was further incorporated into the architecture, which was shown in supplemental Figure S8. Specifically, the architecture was comprised with three convblocks (including a 3×3 convolutional layer followed by a batch normalization layer and a rectified linear unit (ReLU) activation layer) for three different resolutions of the input hyper-images, 8 residual blocks (Resblock), and one fully connected layer. Finally, a softmax activation layer was connected to the last fully connected layer, which was used to yield the prediction probabilities of nodule candidates. Additionally, one dropout layer with probability of 0.5 was added to the fully connected layers.

The training of the model focuses on the optimization of the parameters of the SResCNN model to build a relationship between PET/CT images and EGFR mutation status (positive: 1 or negative: 0). Binary cross entropy was employed as the loss function, while the Adam optimizer was used with an initial learning rate = 0.0001, beta_1=0.9, beta_2=0.999. The learning rate was reduced by a factor of 5 if no improvement of the loss of the validation dataset was seen for a 'patience' number (n=10) of epochs.

The number of the filters, the learning rate and the batch size was determined according the predictive performance on the validation cohort using grid search method.

In order to reduce the risk of overfitting, several techniques were deployed. 1) Augmentation: During the training, augmentation including width/height-shift, horizontal/vertical-flip, rotation and zoom were used to expand the training dataset to improve the ability of the model to generalize. 2) Regularization: L2 regularization was used, which added a cost to the loss function of the network for large weights. As a result, a simpler model that was forced to learn only the relevant patterns in the training data would be obtained. 3) Dropout: Dropout layer, which would randomly set output features of a layer to zero during the training process, was added. 4) Early stop: During training, the model is evaluated on the validation dataset after each epoch. The training was stopped after waiting an additional 30 epochs since the validation loss started to degrade.

Application of the SResCNN network

The generated hyper-image was input into the SResCNN model after z-score normalization, and a deeply learned score (DLS) representing the EGFR mutation positivity could be yielded after a sequential activation of convolution and pooling layers. To develop a robust prediction, all valid slices of each patient were fed into the SResCNN model and the average DLSs with equal weight for each slice was regarded as the final EGFR positive probability of the tumor.

[S1] Kawahara, J., & Hamarneh, Multi-resolution-tract CNN with hybrid pretrained and skin-lesion trained layers. In International workshop on machine learning in medical imaging, 164-171 (2016).

[S2] Zuo, W., et al. Multi-resolution CNN and knowledge transfer for candidate classification in lung nodule detection. IEEE Access, 7, 32510-32521 (2019).

17. The authors should emphasize that an important limitation of their work is the reliance on PET imaging. PET imaging is not available in many parts of the world, and as such this model is going to be limited to academic settings. This also has to be reflected in the title of the manuscript.

Reply:

Thank you for your useful suggestion. Through efforts of the WHO and IAEA, PET imaging is becoming increasingly available in the developing world. However, the WHO has estimated that less than 30% of the world's population has access to any diagnostic imaging tests. Hence, we acknowledge the limitation and have revised the title to **“Non-invasive decision support for NSCLC treatment using PET/CT radiomics”**

And we have also added this limitation in the revised version as follows:

“Lastly, this work is based on PET/CT imaging, which is not widely available in many parts of the world. Therefore, this model may be limited to the developed world and to large urban centers in the developing world.”

18. Another limitation is that the actual EGFR-DLS is picking up quite some variance from standard clinical variables including sex, histology and SUVmax, and potentially other factors that were not tested? E.g. size, PDL1 status? This is a limitation as in essence it can be seen that the imaging predictor is an amalgamation of clinical and molecular data.

Reply:

Thank you for your thoughtful comment. The results of the multivariate linear regression show that the actual EGFR-DLS is picking some variance from clinical variables limited to: sex, histology and SUVmax. However, only 25.0% of EGFR-DLS variability could be explained by these three parameters, which means the current standard clinical variables are not enough to impact the EGFR-DLS. Other clinical variables did not contribute. Further, the clinical signature constructed with these standard clinical variables only achieved AUCs of 0.78, 0.78 and 0.70 in the training, validation and external test cohort, which are significantly poorer compared to the EGFR-DLS with AUCs of 0.86, 0.83, and 0.81 in the training, validation and external test cohort, respectively. There is a potential that some other uncommon factors may be predictive in EGFR mutation status, and highly correlated with EGFR-DLS, which need further investigation. Comparatively, EGFR-DLS could reflect more information in predicting EGFR mutation status in an easier way with the more commonly used

PET/CT images. We have added this limitation in the revised version as follows.

“Fifth, though 25.0% of EGFR-DLS variability could be explained by the amalgamation of some standard clinical variables, EGFR-DLS could reflect more information and achieve significant higher performance in predicting EGFR mutation status in an easier way with the more commonly used PET/CT images.”

Response to REVIEWER #3

The paper has a number of strong features like the use of a really large multi-institutional cohort, a fairly well written paper, with some very nice results regarding the classification of EGFR status using the deep learning model and the association with survival using the deep learning model for classifying EGFR status.

However, there are several aspects in which this paper needs improvement.

1. First and foremost is the methods. There are too few details provided for the methods. For instance, what is the deep learning score? How is this derived? Was it just an average of a bunch of features from a particular layer? How many features were in this layer, if yes?

Reply:

Thank you for your suggestion since addressing this will improve the portability and quality of this work. As noted to our response to reviewer #1, to make the manuscript concise, we apologize for omitting some pertinent details of the deep learning model.

The deep learning score was the output of the deep learning model, and we have stated in the introduction and methods section of the revised version as follows.

“Main

To evaluate the performance of the EGFR prediction model, the interval validation cohort and external test cohort from the Fourth Hospital of Harbin Medical University (HMU), Harbin, China were used. Using the model generated deep learning score (EGFR-DLS), further evaluation of the potential value in guiding therapy choice was performed in the TKI-treated patients from HMU and ICI-treated patients from H. Lee Moffitt Cancer Center & Research Institute (HLM), Tampa, Florida, respectively (Details shown in Figure 1).

Development of the deep learning model

The output of the network, i.e. the deep learning score (EGFR-DLS), was used as the classification result to represent the EGFR mutation positivity probability.”

More details about the deep learning model and the training process in the **Patients and Methods** and **Supplemental methods S2** as described in our response to Comment #16 of the REVIEWER #1.

2. What is the deep learning model used for training? How was the training done? What was the network that was used? Please present details of training, optimization, learning rates, hyper-parameter tuning, network architecture etc. Without this, it’s impossible to understand what these results even mean.

Reply:

The deep learning model was used to predict binary EGFR mutation status with PET/CT images. The inputs of the deep learning model were the ROIs selected by the radiologists that included the whole tumour and surrounding tissue, and the EGFR mutation status (positive or negative) was encoded to one-hot and used as the label. The output of the network, i.e., the deep learning score (EGFR-DLS), was the classification result to represent the probability of EGFR mutation positivity. The details of the training, optimization, learning rates, hyper-parameter tuning, and network architecture have now been added in the Patients and Methods and Supplemental methods S2 as follows.

“Patients and Methods

Development of the deep learning model

*The EGFR mutation status prediction 2D small-residual-convolutional-network (SResCNN) model is presented in **Supplemental Figure S8**. The regions of interest (ROIs) of the PET and CT images were first selected by experienced nuclear medicine radiologists (L.J, JY. Z, and Y.S) after registration using ITK³⁰ on the condition that entire tumor and at least 10 mm of its peripheral region were included, and were then resized to 64x64 pixels by spline interpolation and constructed a three-channel hyper image together with their fusion image (alpha-blending fusion³¹, $\alpha=1$) (Pipeline is shown in **Supplemental Figure S9**). To reduce the effect of the difference between the central slice and peripheral slices, only ROIs that contained measurable tumor tissue were regarded as valid ROIs, and were fed into the SResCNN model to update the parameters of the SResCNN model with backward propagation. The EGFR mutation status (positive/negative) was encoded to one-hot and used as the label. The output of the network, i.e. the deep learning score (EGFR-DLS), was used as the classification result to represent the EGFR mutation positivity probability.*

*EGFR mutation positivity probability at the patient level was obtained by averaging the EGFR-DLSs of the slices that included tumor tissue. To reduce overfitting, augmentation including width/height-shift, horizontal/vertical-flip, rotation and zoom for the 13,583 training hyper-images were used, and the model with the best performance on the validation dataset was selected. Details are shown in **Supplemental S2**.*

“Supplemental S2: Details of the deep learning model

Preparation of the input images

Using ITK-SNAP software, the PET and CT images were firstly registered, and then a square or an irregular box that was close to the boundary of the tumor was delineated, by experienced nuclear medicine radiologist. After resampling, dilation, and resize using cubic spline interpolation, the PET region of interest (ROI) and CT ROI were obtained keeping the entire tumor and its peripheral region with the same size (64x64). Subsequently, the fusion images were calculated through the α -fusion equation:

$$I_{FUSE} = I_{PETnorm} + I_{CTnorm}$$

*where $I_{PETnorm}$ and I_{CTnorm} are the normalized PET and CT pixel-wise image data by z-score normalization. The fusion ROI was further standardized by z-score normalization, and constructed a 3-channel hyper-image together with the normalized PET ROI and normalized CT ROI. This hyper-image was used as the input of the SResCNN model (Pipeline is shown in **Supplemental Figure S9**). Z-score*

normalization means the ROI image was subtracted by the mean intensity value and divided by the standard deviation of the image intensity, before inputting to the deep learning model, to reduce the effect of different equipment and different reconstruction parameters. Because of the big difference of the central slice and peripheral slices, only the slices with the area larger than the 30% of the maximum area of this patient were regarded as valid input images and were used as the input of the deep learning model. The area here means the area of the smallest square including the selected region (Supplemental Fig S9c). Finally, 13,583 training hyper-images were generated for training.

Structure and training of the small-residual-convolutional-network (SResCNN)

The 2D SResCNN is based on several residual blocks with 3-channel input images, which is similar to the well-known Resnet18 network with fewer filters. Given single resolution may not be optimal and depends on the scale of the objects within the image, multi-resolution CNN model was proposed and proved to have significantly better performance [s1, s2]. Therefore, the concept of multi-resolution was further incorporated into the architecture, which was shown in supplemental Figure S8. Specifically, the architecture was comprised with three convblocks (including a 3×3 convolutional layer followed by a batch normalization layer and a rectified linear unit (ReLU) activation layer) for three different resolutions of the input hyper-images, 8 residual blocks (Resblock), and one fully connected layer. Finally, a softmax activation layer was connected to the last fully connected layer, which was used to yield the prediction probabilities of nodule candidates. Additionally, one dropout layer with probability of 0.5 was added to the fully connected layers.

The training of the model focuses on the optimization of the parameters of the SResCNN model to build a relationship between PET/CT images and EGFR mutation status (positive: 1 or negative: 0). Binary cross entropy was employed as the loss function, while the Adam optimizer was used with an initial learning rate = 0.0001, $\beta_1=0.9$, $\beta_2=0.999$. The learning rate was reduced by a factor of 5 if no improvement of the loss of the validation dataset was seen for a 'patience' number ($n=10$) of epochs.

The number of the filters, the learning rate and the batch size was determined according the predictive performance on the validation cohort using grid search method.

In order to reduce the risk of overfitting, several techniques were deployed. 1) Augmentation: During the training, augmentation including width/height-shift, horizontal/vertical-flip, rotation and zoom were used to expand the training dataset to improve the ability of the model to generalize. 2) Regularization: L2 regularization was used, which added a cost to the loss function of the network for large weights. As a result, a simpler model that was forced to learn only the relevant patterns in the training data would be obtained. 3) Dropout: Dropout layer, which would randomly set output features of a layer to zero during the training process, was added. 4) Early stop: During training, the model is evaluated on the validation dataset after each epoch. The training was stopped after waiting an additional 30 epochs since the validation loss started to degrade.

Application of the SResCNN network

The generated hyper-image was input into the SResCNN model after z-score normalization, and a deeply learned score (DLS) representing the EGFR mutation positivity could be yielded after a sequential activation of convolution and pooling layers. To develop a robust prediction, all valid slices of each patient were fed into the SResCNN model and the average DLSs with equal weight for each slice was regarded as the final EGFR positive probability of the tumor.

[S1] Kawahara, J., & Hamarneh, Multi-resolution-tract CNN with hybrid pretrained and skin-lesion trained layers. In International workshop on machine learning in medical imaging, 164-171 (2016).

[S2] Zuo, W., et al. Multi-resolution CNN and knowledge transfer for candidate classification in lung nodule detection. IEEE Access, 7, 32510-32521 (2019).

3. The authors also indicate to two different patterns Pattern I and II, which I presume is based on some form of hierarchical clustering from the Figure 2A. However, these details need to be explained clearly. Again, please explain the clustering algorithm used, how many features were used in the clustering, what layer features were used etc. Were the features from one particular layer or multiple? Which layer features, how this layer was selected and on what dataset was the clustering done etc, how the number of clusters were chosen should be presented.

Reply:

Thank you for your useful suggestion, and we have added concise detail in both the results and methods section as follows.

“Results

Distribution and characteristics of EGFR-DLS

By performing an unsupervised hierarchical clustering on the deeply learned features (i.e., the output of last global average pooling layer, N=256), two patterns were obtained, as shown in Figure 3a. These patterns (I and II) were distinguished by a significantly higher EGFR mutation rate ($p<.001$), proportion of females ($p<.001$), adenocarcinomas ($p<.001$), and never smokers ($p<.001$) in pattern II for both the training and validation cohorts.

Patients and Methods

Visualization of the SResCNN model

Additionally, the deeply learned features (i.e., the output of last global average pooling layer, N=256) were clustered based on the similarities and dissimilarities with unsupervised hierarchical clustering using MATLAB, which was presented by heatmap to show the distinguishable expression pattern among different patients in the training, validation and external HMU test cohorts respectively. In order to investigate the correlation between the different patterns and the EGFR mutation status (positive or negative), two clusters were chosen to be presented.”

Results: The strongest results are the classification of the EGFR status and the comparison to the clinical model and the SUV model. My suggestion would be to put this result up front and provide more details of the models trained for the clinical and SUV models to explain the comparison. Also in addition to just the AUCs please provide other metrics like specificity, sensitivity etc. to get a better understanding of how well the classifier works.

Reply:

Thank you for your suggestions. We have now put these results up front and provided more details of the models for the clinical and combined signature in the results. Figure 2 and Supplemental Table S1 now contain

the accuracy, specificity and sensitivity metrics, which are also now stated in the results as follows.

“Performance of EGFR-DLS in predicting EGFR mutation status

To discriminate EGFR mutant from wild-type, the EGFR-DLS yielded area under the receiver operating characteristics curves (AUCs) of 0.86, 0.83, and 0.81, and accuracies (ACCs) of 81.1%, 82.8% and 78.5% in the training, internal validation, and external test cohorts, respectively (Figure 2 and Supplemental Table S1). These were significantly higher than the commonly used SUVmax, which yielded AUCs of 0.62 ($p<0.001$, Delong test), 0.69 ($p<0.001$, Delong test), and 0.50 ($p<0.001$, Delong test), and ACCs of 58.0% ($p<0.001$, McNemar's test), 72.2% ($p=0.003$, McNemar's test), and 72.2% ($p<0.001$, McNemar's test) in the three cohorts respectively.

When investigating the added value of EGFR-DLS to standard clinical variables (age, sex, stage, histology, Smoking status, and SUVmax), a clinical signature (CS model) was created by combining sex, histology, and smoking status (all other variables were uninformative), and a combined signature incorporating EGFR-DLS, histology and smoking status (CMS model) were built with multivariable logistic regression analysis based on the training cohort. Their quantitative performance shown in Figure 3 and Supplemental Table S1 indicate that the CMS model had the better performance with AUCs of 0.88, 0.88, and 0.84, ACCs of 82.3%, 82.9% and 80.0% in the training, internal validation, and external test cohorts, respectively. These were significantly higher than the CS with AUCs of 0.78 ($p<0.001$, Delong test), 0.78 ($p<0.001$, Delong test), 0.70 ($p=0.005$, Delong test), and ACCs of 72.5% ($p<0.001$, McNemar's test), 72.7% ($p=0.015$, McNemar's test) and 64.6% ($p=0.055$, McNemar's test), respectively. However, the difference between the CMS model and the EGFR-DLS by itself was negligible ($p>0.05$). Additionally, EGFR-DLS was the only identified significant independent variable in EGFR prediction in all three cohorts (Supplemental Table S2).”

Figure 2. Performance of the EGFR-DLS in the EGFR prediction in the different cohorts. The first line shows the ROC curves of different models in the training, validation and HMU test cohorts, respectively. The second line shows the AUC value and the comparison results with Delong-test. *** means p value<.001, ** means p value<.01, * means p value<.05.

Table S1. EGFR predictive performance of different models

	Training cohort	Validation cohort	HMU test cohort
AUC			
SUVmax	0.62 (0.56, 0.67)	0.69 (0.61, 0.77)	0.50 (0.35, 0.65)
CS	0.78 (0.74, 0.82)	0.78 (0.72, 0.84)	0.70 (0.58, 0.82)
EGFR-DLS	0.86 (0.83, 0.90)	0.83 (0.78, 0.89)	0.81 (0.72, 0.92)
CMS	0.88 (0.85, 0.91)	0.88 (0.84, 0.93)	0.84 (0.74, 0.93)
Accuracy			
SUVmax	58.04 (53.85,62.24)	72.19 (65.78,78.07)	53.85 (43.08,64.62)
CS	72.49 (68.07,76.69)	72.73 (66.31,79.14)	64.62 (53.85,76.92)
EGFR-DLS	81.12 (77.39, 4.62)	81.82 (76.22,86.63)	78.46 (68.5, 87.69)
CMS	82.28 (78.79,85.78)	82.89 (77.01,88.24)	80.00 (70.77,89.23)
Sensitivity			
SUVmax	34.33 (27.36,41.29)	52.00 (40, 62.67)	72.22 (58.33,86.11)
CS	78.11 (71.64,83.58)	76.00 (65.33,85.33)	72.22 (58.33,86.11)

EGFR-DLS	84.58 (79.86,89.05)	90.67 (84, 97.33)	69.44 (55.56,83.33)
CMS	83.58 (78.61,88.56)	90.67 (84, 97.33)	69.44 (55.56,83.33)
Specificity			
SUVmax	78.95 (73.68,84.21)	85.71 (78.57,91.96)	31.03 (13.79, 8.28)
CS	67.54 (61.4, 73.68)	70.54 (62.50,78.57)	55.17 (37.93, 2.41)
EGFR-DLS	78.07 (72.81,83.33)	75.89 (67.86,83.93)	89.66 (77.67, 100)
CMS	81.14 (76.11,85.96)	77.68 (69.64,84.82)	93.10 (82.76, 100)

Note. Cutoffs for CS, EGFR-DLS and CMS are 0.5. Cutoff for SUVmax is 5 (according to ROC curves of training cohort). CS is short for clinical signature; CMS is short for combined EGFR-DLS and clinical signature.

4. The next interesting result is the association with survival. It is unclear how the cut off was done to compute the association though. Please explain this in detail. Was this done using median value of the DLS score? Or was ROC analysis done?

Reply:

The median value of the EGFR-DLS from the training cohorts was used as the cut-off, and this has been clarified in the revised version in Statistical analysis section as follows.

“Statistical analysis

*The area under the receiver operating characteristics curve (AUC), accuracy (ACC), specificity (SPEC) and sensitivity (SEN) with cutoff of 0.5 and the 95% confidence interval (CI) by the Delong method¹² were used to assess the ability of EGFR-DLS in discriminating EGFR-mutant and EGFR-wild type. **The median value of the EGFR-DLS from the training cohorts was used as the cut-off.** Performance of the EGFR-DLS was likewise compared with other published clinical characteristics, including smoking status, sex, histologic type³⁵ and PET image-based SUVmax^{36,37} with Delong test.”*

5. Some of the results seem rather speculative. For instance, the result regarding two cancers with hot and cold region filters. From the get go, the radiologists selected the region of interest which very likely did not include a large portion of the background. In that case, it’s unsurprising that the algorithm would indeed find these regions. (1) A more interesting experiment would be to provide the algorithm with a test set that does not include the ROIs and includes a much larger portion of the image and see what the algorithm does. (2) Also if you report SSIM, it’s better to report for all the cases instead of just the two examples.

Reply:

Thank you for your comments and proposed experiment, which sounds very interesting.

(1) ROIs including a much larger portion of the image will need a more complex model to identify more organs and tissues. Additionally, to minimize the effect of losing precision, the larger bounding box means a larger size of ROI for training. Both of the above yields a larger computational burden. In order to design a system that

can be deployed on a local workstation with a few GPUs and be trained with limited dataset, we used the current smaller bounding box that include small portion of the image for training and test in this work. This is because the main aim of this study is to predict the EGFR mutation status of the tumor, the object of this research is the tumor.

Herein, we also went back and identified ROIs with much larger bounding boxes by dilation of the smallest square including the manual delineation with a square of size 60, 80, and 100 mm for 60 cases from the validation cohorts, and the generated EGFR-DLSs achieved AUCs of 0.77 (95%CI: 0.65,0.89) ($p=0.38$, Delong test), 0.70 (95%CI: 0.56, 0.83) ($p=0.11$, Delong test), and 0.63 (95%CI: 0.49, 0.78) ($p=0.04$, Delong test), which are smaller compared to the original AUC of 0.83 (95%CI: 0.73,0.93). With the input of a ROI including more organs and tissues, the prediction ability was decreased. This was reasonable, since the training of the model was based on the ROI with 10~20 mm of tumor peripheral region included, without considering the situation with more organs and tissues included. A model that works for a larger ROI with more organs and tissues should be trained with larger ROIs based on more data. Though smaller AUCs were obtained with more organs and tissues included, the difference was not statistically different for the dilation with square of size 60 mm. For this dilation size, we visualized the model using larger ROIs of the same two patients shown in Figure 3 as shown in the following figure (Figure S1). From this figure, with the input of a ROI including more organs and tissues, the shape and the location of the activation maps and the positive/negative filters were not the same, but they are very similar to the original. We have also clarified this in the revised manuscript as follows.

“Distribution and characteristics of EGFR-DLS

“Similarly, the negative filter was strong activated and the positive filter was nearly shut down with EGFR-wild-type tumor fed to the deep learning model, which reveals the strong classification ability of the deep learning model. When the input ROIs were enlarged to include more organs and tissues, similar activation maps, positive/negative filters and predicted EGFR-DLSs were also obtained as shown in Supplemental Figure S1.”

Unfortunately, given this model is trained mainly for the tumor region, we are afraid that it shouldn't be used with the input of the ROI without tumor region. This will be our future study that developing a model with both high tumor location ability and high EGFR prediction ability. We have added this in the limitation section as follows:

“Seventh, given this model is trained mainly for the tumor with 10~20 mm of tumor peripheral region included, the model couldn't be used for the ROIs without tumor included, and the prediction ability will be decreased with the input of ROI including more organs and tissues. A more intelligent model to solve this problem will be left for our future work.”

Figure S1. **Visualization of the model using different ROIs of the same patients in Figure 3.** The first lines are the original input ROIs, and the second line show the two of the activation maps of the fourth ResBlocks, the positive filter and the negative filter generated with the original input ROI. The third and fourth lines are the input ROIs with more organs/tissues included, and the corresponding activation maps and positive/negative filter.

2) The SSIM was reported with the median value with interquartile range (IQR) for all cases as follows.

“Further, the hot-spot regions shown in negative and positive filters (Figure 3c-d, row 3, columns 3,4) also corresponded well with the ^{18}F -MPG uptake of the EGFR wild type and mutant type with a median structural similarity Index (SSIM)²¹ of 0.66 (interquartile range (IQR): 0.38, 0.77) (0.66 for Figure 3c and 0.70 for Figure 3d, respectively).”

6. The result regarding the radiologist intra correlation coefficient study is rather confusing. What is the objective here? What is the ICC computed for. Please explain this.

Reply:

Thank you for your useful suggestion. In this work, the rough ROIs were selected by radiologist using ITK software (Pipeline is shown in Supplemental Figure S9). However, given there were minor differences among different radiologists in selecting the rough ROIs, ROIs of 73 patients within the validation cohort were selected by all three radiologists to validate the reproducibility of DLS. The inter-rater agreement of DLS estimations were calculated by intraclass correlation coefficient (ICC) among the three radiologists. We have added concise detail in both the results and methods section as described in Comment #10 of REVIEWER #1.

7. The results regarding the reporting of correlations need to be toned down a bit. There doesn't seem to be any strong correlations. The highest correlation is around 0.48 which is considered moderate. A lot of others even if the Spearman P value is significant are weak to low. Please report them as such. Lack of association (e.g. $P=0.068$) should not be reported as higher.

Reply:

Thank you for your useful suggestion, and we have revised the related description of the relationship as follows.

*“For the patients with consistent results between EGFR status from biopsy and 18F-MPG imaging (N=64), the EGFR-DLS derived from 18F-FDG was **moderately** positively correlated with 18F-MPG accumulation in tumors measured by 18F-MPG SUVmax (Spearman $\rho=0.48$, $p<.001$, Figure 4a).*

*For the patients with known PD-L1 expression, a **weak** but significant inverse correlation was observed between the PD-L1 status and EGFR-DLS with Spearman’s ρ of -0.24 ($p<.001$), -0.26 ($p=0.006$), and -0.26 ($p=0.024$) for the training, validation, and HLM ICI-treated sub-cohorts, respectively (Supplemental Figure S4).*

*Further, if you grouped patients based on the 40 patients with controlled disease (SD/PR/CR) the EGFR-DLS was higher (median: 0.52) **though not significant**, compared to the 27 patients with PD (median: 0.38) (Wilcoxon’s $p=0.068$).”*

Response to REVIEWER #4

The proposed approach is novel and very interesting however there are some points that need to be clarified and discussed. More specifically,

1. The title of the manuscript suggests the development of a treatment decision support system based on retrospective data (observational rather than interventional approach) which raises questions since hidden colliders might kick-in and hamper the clinical value of the proposed DSS. In particular recent publications (1,2) indicate the necessity to incorporate causal inference in order to estimate the effect of a treatment selection on clinical outcomes when using observational data. Potential hidden colliders like patient fitness, age, ethnicity and others need to be considered in order to reduce selection bias that is inherent to the observational data used in the current study. I would strongly suggest that a relative text should be discussed in the paper, providing arguments why such an approach was not followed.

References

1. van Amsterdam WAC, Verhoeff JJC, de Jong PA, Leiner T, Eijkemans MJC. Eliminating biasing signals in lung cancer images for prognosis predictions with deep learning. NPJ Digit Med. 2019;2:122. Published 2019 Dec 10. doi:10.1038/s41746-019-0194-x
2. Paul W Holland. Statistics and causal inference. Journal of the American statistical Association, 81(396):945-960, 1986.

Reply:

Thank you for your suggestion, which sounds very interesting. Sex, ethnicity and others may indeed introduce selection bias. The first referred publication (Amsterdam *et. al.*) provided a very novel and useful way to solve the selection bias problem and we have cited it here. However, this is a retrospective study, and the training of the model was limited in the current data. Not all the data have these colliders and clinical outcome at the same time. For example, the HLM cohort doesn't have the EGFR mutation status, while the SPH and HBMU cohorts don't have the clinical outcome of TKI-treatment or ICI-treatment. Therefore, we could not follow this approach in this study, but it will be left for future work. Additionally, the results of the test cohorts with different demographic characteristics (e.g. different ethnicities, different histology) compared to the training cohorts supports the generalization of the model, indicating that this model was less affected by the hidden colliders. We have stated this in the limitation section as follows.

“Sixth, the hidden colliders like sex, ethnicity and histology may introduce the selection bias in the current study. Though CNN model with causal inference incorporated provided a good way to reduce this bias²⁷, not all the patients have the information of these colliders and clinical outcome at the same time. For example, the HLM cohort doesn't have the EGFR mutation status, while the SPH and HBMU cohorts don't have the clinical outcome of TKI-treatment or ICI-treatment. Therefore, this method will be left for future work. Additionally, the satisfied results of the test cohorts with different demographic characteristics (e.g. different ethnicities, different histology) further validated that the model was less affected by the hidden colliders.”

2. The proposed deep neural network is comprising of almost 1.4 million trainable parameters which is way more than the available samples for training (9911 training patches). The latter raises concerns for overfitting and these needs again to be discussed in a way to provide arguments on how the authors prevented overfitting using such heavyweight architecture. Also it needs to be very clear that hyper images of the same patient didn't participated in the various data sets simultaneously (training, validation, test).

Reply:

Thank you for your comments. We apologize for the error in the previous version; the correct number of ROIs is 13,583 ROIs for training. Though the dataset was not big enough, several techniques were deployed to reduce overfitting. We have clarified in the supplemental S2 as follows:

“Structure and training of the SResCNN network

In order to reduce the risk of overfitting, several techniques were deployed. 1) Augmentation: During the training, augmentation including width/height-shift, horizontal/vertical-flip, rotation and zoom were used to expand the training dataset to improve the ability of the model to generalize. 2) Regularization: L2 regularization was used, which added a cost to the loss function of the network for large weights. As a result, a simpler model that was forced to learn only the relevant patterns in the training data would be obtained. 3) Dropout: Dropout layer, which would randomly set output features of a layer to zero during the training process, was added. 4) Early stop: During training, the model is evaluated on the validation dataset after each epoch. The training was stopped after waiting an additional 30 epochs since the validation loss started to degrade. “

Additionally, the hyper images for training, validation and test were generated based on the patient level. And we have clarified this in the methods section as follows:

“Patients and Methods

Study population

Patient cohorts from SPH and HBMU were divided into a training (n=429) and validation cohort (n=187) randomly with a ratio of 70/30 to train and validate the deep learning model to predict EGFR mutation, while patients from HMU were used as external test cohort to test this model. Data from cohorts was rigorously kept separate. “

3. The proposed architecture is 2d, which means that the probabilities for predicting EGFR status is on a slice basis rather than a patient or a tumor basis. a) How the authors aggregated these probabilities to transform them on a per tumor prediction? Please clarify. b) The selected architecture also needs some justification, why the authors selected the specific one? c) Why they didn't consider more suitable approaches including pre-trained networks with transfer learning given the typical $n \ll p$ situation they had?

Reply:

Thank you for your comments.

- a) To develop a robust prediction, the average EGFR-DLSs of all valid slices that included tumor tissue fed into the SResCNN model with equal weight was regarded as the final EGFR positive probability of the tumor. This has been re-stated in the methods section as follows.

“Patients and Methods

Development of the deep learning model

To reduce the effect of the difference between the central slice and peripheral slices, only the ROIs contained measurable tumor tissue were regarded as valid ROIs.

EGFR mutation positivity probability at the patient level was obtained by averaging the EGFR-DLSs of the valid slices that included tumor tissue. “

- b) Resnet is one of the best architectures for the image classification. Given the limited number of the datasets, we constructed a network similar to Resnet-18 but with fewer filters, which could be trained on our own workstation with less data. Additionally, given single resolution may not be optimal and depends on the scale of the objects within the image, multi-resolution CNN model was proposed and proved to have significantly better performance [s1,s2]. Therefore, we incorporated the concept of multi-resolution into the smaller Resnet structure. By trying different number of filters, the current structure could give an acceptable result. We have clarified in the supplemental S2 as follows:

“Structure and training of the SResCNN network

The 2D SResCNN is based on several residual blocks with 3-channel input images, which is similar to the well-known Resnet18 network with fewer filters. Given single resolution may not be optimal and depends on the scale of the objects within the image, multi-resolution CNN model was proposed and proved to have significantly better performance [s1, s2]. Therefore, the concept of multi-resolution was further incorporated into the architecture, which was shown in supplemental Figure S8.

The number of the filters, the learning rate and the batch size was determined according the predictive performance on the validation cohort using grid search method.”

[S1] Kawahara, J., & Hamarneh, Multi-resolution-tract CNN with hybrid pretrained and skin-lesion trained layers. In International workshop on machine learning in medical imaging, 164-171 (2016).

[S2] Zuo, W., et al. Multi-resolution CNN and knowledge transfer for candidate classification in lung nodule detection. IEEE Access, 7, 32510-32521 (2019).

- c) First, we did try transfer learning based on Resnet-50 using Keras, which has been used in many studies. However, it didn't result in any improvements over the model that we currently used (it actually performed worse). Second, we want to reveal the biological meaning of the model through the visualization. Given the pre-trained models were trained with real-world images and might extract real-world-related features that might not applicable in tumor images, we feel that pre-trained models might not give better explanation of the features of the middle layers.

REVIEWER COMMENTS

Reviewer #1 (Remarks to the Author):

Minor comment:

- Please change “univariable” to “univariate” in Table S2 and Table S5.

Reviewer #3 (Remarks to the Author):

The manuscript is much improved from the previous version. Its also nice that the authors provide the code and the model used in this work for reproducible research.

There are however a few outstanding questions regarding the method:

1. What type of image registration was used to fuse the CT and PET images? This is important because, poor registration would basically diminish the accuracy of the fused PET/CT images. Weren't the PET/CT scans acquired as a single acquisition? If a contrast CT scan from a different time was fused with a PET scan, this is a much more difficult image registration and its not clear if the standard ITK-SNAP methods would allow any decent registration. Please provide more details.

2. The authors describe a image resizing, resampling, and dilation step. Can you please provide more details regarding this? What type of image resampling was used for instance? Depending on the resampling technique, you can get less accurate results. Also, what is the rationale for dilation here and what is getting dilated? There is no tumor contour -- so its not clear what the dilation step means here.

3. What do the authors mean by z-score normalization? Z-score normalization inside the ROI based on the ROI mean and SD? Please clarify. Why is this better than standard rescaling of intensities? Wouldn't this really depend on the ROI and the extent of tumor within the ROI. So did the radiologists carefully select the ROI so that there was always a similar proportion of air/tumor/normal tissue etc? Also how does this work depending on tumor location - say tumors located in the mediastinum vs. tumors surrounded by air vs. tumors attached to chestwall?

4. The deep learning architecture description is somewhat confusing. It is claimed that a multi-resolution architecture was used but with fewer parameters than ResNet18. How many multiple resolutions did you use? Also, if you add additional resolution inputs, you automatically increase the number of parameters. Doesn't that defeat the purpose of using fewer convolutional layers. Would then not having more convolutional layers to extract features at different scales be better than using different resolution inputs? Also, how was the number of resolutions, and the actual number of resolution images (this is not specified) arrived at? These are all part of an ablation test and these should be included in the supplements. It looks like some of this analysis was done to arrive at the right architecture.

Related to this, the authors say that ResNet18 didn't work (or at least that's how I understood it as the rationale for using a similar network with fewer layers). Presenting comparison results of the

actual ResNet18 would be helpful.

Was the network fine-tuned or trained from scratch? Please provide details.

The authors use a fused PET-CT image as a third channel. Of the three channels, is there a sense of which image contributes or is most useful? I think an ablation experiment removing one of the image channels at a time, (you can for example use CT images only in all 3 channels, or PET only, or PET-CT fused only). The reason this experiment would be interesting is because if it turns out that CT alone gets you similar accuracy as the PET-CT, then you can use the network with CT only images, which is more generally applicable. On the other hand, if there is a clear advantage of using PET-CT, that would be good to know as well.

Reviewer #4 (Remarks to the Author):

All comments were addressed satisfactory by the authors, my recommendation is to publish the manuscript.

Response to REVIEWER #1

1. Minor comment:

- Please change “univariable” to “univariate” in Table S2 and Table S5.?

Reply:

Thank you for your correction. We have now changed “univariable” to “univariate” in Table S2 and Table S5.

Response to REVIEWER #3

1. What type of image registration was used to fuse the CT and PET images? This is important because, poor registration would basically diminish the accuracy of the fused PET/CT images. Weren't the PET/CT scans acquired as a single acquisition? If a contrast CT scan from a different time was fused with a PET scan, this is a much more difficult image registration and its not clear if the standard ITK-SNAP methods would allow any decent registration. Please provide more details.

Reply:

Thank you for your thoughtful comments, and we apologize for the lack of clarity.

For all cases in this study, contemporaneous planar CT images were acquired and applied to the image registration with PET images. The SOPs from the institutions that conducted the PET/CT imaging studies first acquired a non-contrast CT scan, and then the PET scan was acquired next. The PET and CT images were co-registered on the same machine by scanner software. Thus, most of the patients in this analysis were already registered well prior to our analysis. We acknowledge that a few patients may have minor misalignment due to respiratory motion as the CT image acquisition is rapid whereas the PET acquisition requires much longer meaning that the CT is a snapshot of position while the PET is more of an integral of average position. For the small number of patients of mis-registration, a rigid transformation was performed using ITK-SNAP by an experienced nuclear medicine radiologist in the manual mode to fuse the non-contrast CT and PET images. We have now clarified this in the supplemental S2 as follows.

“Pipeline of input images generation is shown in Supplemental Figure S9. In clinical practice, a non-contrast CT scan was acquired first and then PET scan was acquired subsequent. The PET and CT images were co-registered on the same machine by scanner software. Thus, almost all cases of this study are already registered. A few cases had minor misalignment due to respiratory motion. For these cases, an experienced nuclear medicine radiologist manually adjusted the alignment using ITK-SNAP. Then a square or an irregular box that was close to the boundary of the tumor was delineated using ITK-SNAP software by the experienced nuclear medicine radiologist.”

2. The authors describe an image resizing, resampling, and dilation step. Can you please provide more details regarding this? What type of image resampling was used for instance? Depending on the resampling technique, you can get less accurate results. Also, what is the rationale for dilation here and what is getting dilated? There is no tumor contour -- so its not clear what the dilation step means here.

Reply:

We agree that resampling may introduce some errors that may lead to less accurate results, but this should not be a differential bias/error in relation to the outcome/dependent variables. Nonetheless, to ensure the PET and CT images with the same resolution and that the input ROI have the same size, resampling is necessary. In our work, bicubic interpolation was used for image resampling since it can get relatively clear picture quality and is most commonly used in image processing compared to other resampling techniques [s1].

Regarding the dilation, though there is no tumor contour, a square or an irregular box that was close to the boundary of the tumor was delineated by the experienced nuclear medicine radiologist. Then the smallest square mask that included the selected region was obtained, and the dilation was performed on this mask. Through the dilation, the entire tumor and at least 10 mm of the peritumoral region could be included in the input ROI.

We have now clarified this, provided a pipeline of this process and cited in the beginning of supplemental S2 as follows.

“Pipeline is shown in Supplemental Figure S9. In clinical practice, Then a square or an irregular box that was close to the boundary of the tumor was delineated using ITK-SNAP software by the experienced nuclear medicine radiologist. After resampling with bicubic spline interpolation to ensure the PET and CT images have the same resolution, dilation of the smallest square mask including the selected region to keep the peritumoral region included, and resize using bicubic spline interpolation, the PET region of interest (ROI) and CT ROI were obtained keeping the entire tumor and its peripheral region with the same size (64×64).”

Figure S9. **Illustration of the generation of the input hyper-image.** A square or an irregular box, which was close to the boundary of the tumor, was delineated manually in ITK software firstly, and then the hyper-image was generated after resampling (bicubic interpolation), dilation and fusion automatically.

[s1] HAN, Dianyuan. Comparison of commonly used image interpolation methods. In: Proceedings of the 2nd international conference on computer science and electronics engineering. Atlantis Press, 2013.

3. a). What do the authors mean by z-score normalization? Z-score normalization inside the ROI based on the ROI mean and SD? Please clarify. b). Why is this better than standard rescaling of intensities? c). Wouldn't this really depend on the ROI and the extent of tumor within the ROI. d). So did the radiologists carefully select the ROI so that there was always a similar proportion of air/tumor/normal tissue etc? e).

Also how does this work depending on tumor location - say tumors located in the mediastinum vs. tumors surrounded by air vs. tumors attached to chestwall?

Reply:

a). What we meant by Z-score normalization, was that the ROI image was subtracted by the mean intensity value and divided by the standard deviation of the ROI image intensity. We have now clarified this in the supplemental S2 as follows.

" $I_{PETnorm}$ and I_{CTnorm} are the normalized PET and CT pixel-wise image data by z-score normalization, which means that the ROI image was subtracted by the mean intensity value and divided by the standard deviation of the ROI image intensity."

b). Z-score normalization is a widely used normalization technique to eliminate the offset effect [S1] Because it is reported to lead to a smoother and faster convergence in the training of CNN networks [S2, S3], we decided to use Z-score normalization in our work.

c). In response, we performed a further experiment on ROIs with much larger bounding boxes by dilation of the smallest square including the manual delineation with a square of size 60 mm for 60 cases from the validation cohorts, and the generated EGFR-DLSs achieved AUCs of 0.77 (95%CI: 0.65, 0.89) ($p=0.38$, Delong test) which is slightly lower, but not significantly, compared to the original AUC of 0.83 (95%CI: 0.73, 0.93). That is to say, this method could also work with the input of a larger ROI that included more organ/tissue.

d). For the ROI selection shown in supplemental Figure S9, a square or an irregular box, which was close to the boundary of the tumor, was delineated. Therefore, the radiologists did not carefully select the ROI to keep a similar proportion of air/tumor/normal tissue as this would have been impractical. To have a method that is more generally applicable, it was important that we put no constraints on our radiologist regarding choice of input conditions.

e). According to the tumor location, lung cancer could be divided into the following four types: I) tumors surrounded by air; II) tumors surrounded by air and mediastinum; III) tumors surrounded by air and chestwall; IV) tumors surrounded by air, mediastinum and chestwall. In this study, tumor location was not the inclusion or exclusion criteria. Therefore, all different types of tumors were included in this analysis. In detail, for the 65 patients from the external HMU test cohort, 15 patients with an EGFR mutant prevalence of 60.0% belong to Type I, and the EGFR-DLS achieved an AUC of 0.98 (95%CI:0.93-1.00, $p=0.002$); 23 patients with an EGFR prevalence of 34.78% belong to Type II, and the EGFR-DLS achieved an AUC of 0.80 (95%CI: 0.61-1.00, $p=0.020$); 13 patients with an EGFR prevalence of 61.5% belong to Type III, and the EGFR-DLS achieved an AUC of 0.93 (95%CI:0.77-1.00, $p=0.013$); the remaining 14 patients with an EGFR mutant prevalence of 78.6% belong to Type IV, and the EGFR-DLS achieved an AUC of 0.97 (95%CI:0.88-1.00, $p=0.016$). There are no significant difference of the AUC between different types (Type I vs Type II, $p=0.10$; Type I vs Type III, $p=0.53$; Type I vs Type IV, $p=0.82$; Type II vs Type III, $p=0.35$; Type II vs Type IV, $p=0.14$; Type III vs Type IV, $p=0.64$). Therefore, this work is independent on tumor location. And we have added this in the Results section as follows.

"Distribution and characteristics of EGFR-DLS

The intraclass correlation coefficient (ICC) of these three EGFR-DLSs was 0.91 (95%CI:0.87-0.94, $p<.001$), and indicating that there were no significant differences in AUCs of these three EGFR-DLSs (Supplemental Figure S2), both of which validate the reproducibility of EGFR-DLS with the input images selected by different radiologists. Further stratified analysis was also performed to investigate the independence of the model on tumor location. For the external HMU test cohort, the EGFR-DLS achieved AUCs of 0.98 (95%CI:0.93-1.00, $p=0.002$), 0.80 (95%CI: 0.61-1.00, $p=0.020$), 0.93 (95%CI:0.77-1.00, $p=0.013$), 0.97 (95%CI:0.88-1.00, $p=0.016$) in tumors surrounded by air ($n = 15$ cases), tumors

surrounded by air and mediastinum (n = 23 cases), tumors surrounded by air and chestwall (n = 13 cases), and tumors surrounded by air, mediastinum and chestwall (n = 14 cases), respectively. There are no significant difference of the AUC between any two different types ($p=0.10-0.82$), which suggests this work is independent on tumor location."

[S1] Oh, Shu Lih, et al. Automated diagnosis of arrhythmia using combination of CNN and LSTM techniques with variable length heart beats. *Comput Bio Med*, 102, 278-287 (2018).

[S2] Hennrich, J., et al. Investigating deep learning for fNIRS based BCI. In: 2015 37th Annual international conference of the IEEE Engineering in Medicine and Biology Society (EMBC), 2844-2847 (2015).

[S3] Passalis, N., & Tefas, A. Dimensionality reduction using similarity-induced embeddings. *IEEE Trans. Neural Netw. Learn. Syst.*, 29, 3429-3441 (2017).

4. The deep learning architecture description is somewhat confusing. It is claimed that a multi-resolution architecture was used but with fewer parameters than ResNet18. How many multiple resolutions did you use? Also, if you add additional resolution inputs, you automatically increase the number of parameters. Doesn't that defeat the purpose of using fewer convolutional layers. Would then not having more convolutional layers to extract features at different scales be better than using different resolution inputs? Also, how was the number of resolutions, and the actual number of resolution images (this is not specified) arrived at? These are all part of an ablation test and these should be included in the supplements. It looks like some of this analysis was done to arrive at the right architecture.

Related to this, the authors say that ResNet18 didn't work (or at least that's how I understood it as the rationale for using a similar network with fewer layers). Presenting comparison results of the actual ResNet18 would be helpful.

Was the network fine-tuned or trained from scratch? Please provide details.

Reply:

Thank you for your thoughtful comments, and we apologize for the lack of clarity.

The current architecture has the same number of layers as ResNet18. However, the number of filters is much fewer (1/8) for each layer compared to ResNet18. The main reason we can use a smaller number of filters was that we are tasked with only 2 classes to predict compared to the 1000 classes in imagenet.

In this work, we used 3 different resolutions with image sizes of 64x64, 32x32 and 16x16. Though with 3 resolutions, the number of convolutional layers was not added. And the number of filters was still much smaller compared to ResNet18. According to [s1,s2], a multi-resolution CNN model proved to have significantly better performance. Therefore, we only utilized different resolution inputs, and the comparison with multi-scales could be considered for future research. We have prepared different resolutions with image sizes of 64x64, 32x32, 16x16 and 8x8. The actual number was determined according to the predictive performance on the validation cohort using the grid search method, and we have clarified this in supplemental S2 as follows.

"The number of the filters, the number of resolutions, the learning rate, and the batch size was determined according to the predictive performance on the validation cohort using the grid search method."

In response to the comment, we have performed two extra experiments.

1) To validate the necessity of multi-resolution and define the number of resolutions, architectures with different number of resolutions (Supplemental Figure S10) were trained, validated and tested on the conditions that the structures of other convolutional layers were kept the same. The predictive performance of different architectures is provided in supplemental Figure S11. When the number of resolution was less than 4, the predictive performance was improved in the training and validation cohorts with the increase number of different resolutions. When the number of resolution was 4, though predictive performance was improved in the training cohort, no improvement was found for validation cohort. Thus, to keep the architecture with fewer parameters, we used 3 different resolutions in this analysis. Additionally, the advantage of multi-resolution architecture was also proved in the external test cohort, which has more advanced stage cases with larger tumor volume. That is to say, the multi-resolution architecture is more independent on the scale of the objects within the image.

Figure S11. Comparison of different architectures based on predictive performance. 1-resolution means the model with only 1 resolution with image size of 64x64; 2-resolution means the model with 2 resolutions with image size of 64x64, and 32x32; 3-resolution means the proposed model in this work with 3 different resolutions with image size of 64x64, 32x32 and 16x16; and 4-resolution means the model with 4 resolutions with image size of 64x64, 32x32, 16x16 and 8x8. Detailed architectures were provided in supplemental Figure S10.

2) In this study, we believe it is imperative to reveal the biological underpinnings of the model through the visualization. We were concerned that pre-trained models with transfer learning might not yield better explanation of the features of the middle layers since the pre-trained models were trained with real-world images and might extract real-world-related features that might not applicable in tumor images. Additionally, we also tried transfer learning, and significantly worse performance was obtained with AUCs of 0.76 and 0.67 in the training and validation cohorts ($p < 0.05$), respectively. Therefore, compared to transfer learning, we

prefer to construct a network, which could be trained on our own workstation with the actual tumor data.

The ResNet18 trained with our tumor images achieved AUCs of 0.88 (95%CI: 0.85, 0.91), 0.78 (95%CI: 0.72, 0.85) and 0.70 (95%CI: 0.57, 0.83) in the training, validation and external test cohorts, respectively. Compared to 1-resolution model with AUCs of 0.84 (95%CI: 0.80, 0.87), 0.78 (95%CI:0.72, 0.85), and 0.71 (95%CI: 0.59, 0.84) in the training, validation and external test cohorts, respectively, which has similar architecture with ResNet18 but fewer filters for each layer, the performance of ResNet18 is better than in the training cohort, but nearly the same in the validation cohorts and little worse in the test cohort (Supplemental Figures S11). This result indicates that larger number of filters wouldn't increase the predictive performance but increase the risk of overfitting for the task in our study. Additionally, during the training of Resnet18, it took 200 seconds for training each epoch, which is ten times longer than training the proposed model. Therefore, the ResNet18-similar architecture with fewer filters for each layer is more appropriate in our study.

The determination of this architecture has now been provided in Supplemental S3 as follows.

“Supplemental S3: Determination of architectures of CNN network

To select the optimal architecture, we first compared the ResNet18 with a similar architecture but smaller number of filters for each layer (referred as 1-resolution model later, shown in Supplemental Figure S10 (a)). Trained with our tumor images, the ResNet18 achieved AUCs of 0.88 (95%CI: 0.85, 0.91), 0.78 (95%CI: 0.72, 0.85) and 0.70 (95%CI: 0.57, 0.83) in the training, validation and external test cohorts, respectively. Compared to the 1-resolution model (AUC: training: 0.84, 95%CI: 0.80, 0.87; validation: 0.78, 95%CI: 0.72, 0.85; test: 0.71, 95%CI: 0.59, 0.84), the AUC is higher in the training cohort, but nearly the same in the validation cohorts and slightly lower in the test cohort (Supplemental Figures S11). This suggests that greater number of filters would not increase the predictive performance, but does increase the risk of overfitting for the task in our study. Additionally, during the training of Resnet18, it took 200 seconds for training each epoch, which was ten times longer than training the proposed model. Therefore, the ResNet18-similar architecture with fewer filters for each layer is more appropriate in our study.

Further, to validate the necessity of multi-resolution and determine the number of resolutions, architectures with different number of resolutions (Supplemental Figure S10) were trained and tested on the conditions that the structures of other convolutional layers were kept the same. From the performance shown in supplemental Figure S11, when the number of resolution was less than 4, the predictive performance was improved in the training and validation cohorts with the increase number of different resolutions. When the number of resolution was 4, though predictive performance was improved in the training cohort, no improvement was found for validation cohort. In order to keep the architecture with fewer parameters, we used 3 different resolutions in this work. Additionally, the advantage of multi-resolution architecture was also proved in the external test cohort, which has more advanced stage cases wither larger tumor volume. That is to say, the multi-resolution architecture is more independent on the scale of the objects within the image.

Based on the above comparison, the current SResCNN network was arrived at and used in this work.”

[S1] Kawahara, J., & Hamarneh, Multi-resolution-tract CNN with hybrid pretrained and skin-lesion trained layers. In International workshop on machine learning in medical imaging, 164-171 (2016).

[S2] Zuo, W., et al. Multi-resolution CNN and knowledge transfer for candidate classification in lung nodule detection. IEEE Access, 7, 32510-32521 (2019).

5. The authors use a fused PET-CT image as a third channel. Of the three channels, is there a sense of which

image contributes or is most useful? I think an ablation experiment removing one of the image channels at a time, (you can for example use CT images only in all 3 channels, or PET only, or PET-CT fused only). The reason this experiment would be interesting is because if it turns out that CT alone gets you similar accuracy as the PET-CT, then you can use the network with CT only images, which is more generally applicable. On the other hand, if there is a clear advantage of using PET-CT, that would be good to know as well.

Reply:

Thank you for your comments and proposed experiment. In response to your comment, we have performed this experiment, but didn't provide it in the manuscript due to the worse performance compared to the current one. In this version, we have added the corresponding results in the discussion, as follows:

“We also observed that hyper-image constructed with different modalities could significantly improve the accuracy of EGFR mutation modelling. By training similar network only using PET and CT images, the resulting EGFR-DLSs achieved AUCs of 0.76 (95%CI: 0.72, 0.81) and 0.80 (95%CI: 0.76, 0.84) in the training cohort, 0.74 (95%CI: 0.67, 0.81) and 0.75 (95%CI: 0.67, 0.81) in the validation cohort, respectively, which was significantly worse ($p < 0.05$) than those generated using the hyper-images. The similar network with input of PET-CT fused image achieved a lower though not significant AUCs of 0.85 (95%CI: 0.81, 0.88) and 0.79 (95%CI: 0.73, 0.86) in the training ($p = 0.19$) and validation ($p = 0.13$) cohort, respectively. This may be attributed to the important regions used for the accurate prediction of EGFR mutation could be better and easier localized by utilizing both metabolic and anatomical information as reflected by PET and CT images, respectively.”

REVIEWERS' COMMENTS

Reviewer #3 (Remarks to the Author):

The authors have addressed all my comments and concerns. The paper has been improved tremendously and this is a truly exciting work! Congratulations and thank you for addressing the suggestions.